# The Structure of Evolutionary Model Space for Proteins across the Tree of Life

**DOI:** 10.3390/biology12020282

**Published:** 2023-02-10

**Authors:** Gabrielle E. Scolaro, Edward L. Braun

**Affiliations:** Department of Biology, University of Florida, Gainesville, FL 32611, USA

**Keywords:** molecular evolution, substitution matrix, amino acid exchangeability, models of sequence evolution, protein evolution, archaea, bacteria

## Abstract

**Simple Summary:**

The relative rates of amino acid substitution over evolutionary time reflect the chemical properties of amino acids. Substitutions that result in an amino acid similar to an ancestral residue accumulate more rapidly than those resulting in a dissimilar amino acid. The substitution rates for each amino acid pair are the parameters in models of evolutionary change for proteins. Although the best-fitting model of protein evolution is known to differ among taxa, a comprehensive picture of model changes across the tree of life is not available. In principle, models of protein change might reflect evolutionary history (i.e., closely related taxa have similar models) or the environment (i.e., taxa living in similar environments have similar models). We estimated models of amino acid evolution for organisms across the tree of life, finding evidence that history and the environment have both contributed to model differences. Bacterial models differed from archaeal and eukaryotic models. Models for Halobacteriaceae (archaea that live in highly saline environments) and Thermoprotei (a group of thermophilic archaea) were found to be very distinctive. The rates of substitution for pairs of aromatic amino acids were especially variable. Overall, these results paint a picture of the “evolutionary model space” for proteins across the tree of life.

**Abstract:**

The factors that determine the relative rates of amino acid substitution during protein evolution are complex and known to vary among taxa. We estimated relative exchangeabilities for pairs of amino acids from clades spread across the tree of life and assessed the historical signal in the distances among these clade-specific models. We separately trained these models on collections of arbitrarily selected protein alignments and on ribosomal protein alignments. In both cases, we found a clear separation between the models trained using multiple sequence alignments from bacterial clades and the models trained on archaeal and eukaryotic data. We assessed the predictive power of our novel clade-specific models of sequence evolution by asking whether fit to the models could be used to identify the source of multiple sequence alignments. Model fit was generally able to correctly classify protein alignments at the level of domain (bacterial versus archaeal), but the accuracy of classification at finer scales was much lower. The only exceptions to this were the relatively high classification accuracy for two archaeal lineages: Halobacteriaceae and Thermoprotei. Genomic GC content had a modest impact on relative exchangeabilities despite having a large impact on amino acid frequencies. Relative exchangeabilities involving aromatic residues exhibited the largest differences among models. There were a small number of exchangeabilities that exhibited large differences in comparisons among major clades and between generalized models and ribosomal protein models. Taken as a whole, these results reveal that a small number of relative exchangeabilities are responsible for much of the structure of the “model space” for protein sequence evolution. The clade-specific models we generated may be useful tools for protein phylogenetics, and the structure of evolutionary model space that they revealed has implications for phylogenomic inference across the tree of life.

## 1. Introduction

The fact that rates at which different pairs of amino acids experience substitutions over evolutionary time vary by orders of magnitude has been appreciated for more than five decades [1,2]. Indeed, Kimura and Ohta [3] included the fundamental pattern observed in those pioneering studies, in which the rate conservative substitutions (exchanges that involve pairs of chemically similar amino acids) is higher than the rate of more radical substitutions, as one of the five principles governing molecular evolution. The processes that determine the rates of substitution for the various pairs of amino acids can be divided into two fundamental categories: (1) the rate and spectrum of non-synonymous mutations and (2) the probability that any of those novel non-synonymous changes will increase in frequency to the point that they will be observed as substitutions. The rapid accumulation of sequence data in the genomic and post-genomic eras [4,5] has served to confirm the fundamental patterns observed in those classic studies. However, the explosive growth of sequence databases has provided enough information to show differences among clades in their patterns of amino acid substitution [6,7,8,9]. Those more recent results indicate that the processes governing the relative rates of amino acid change have themselves changed over evolutionary time.

Understanding the ways that patterns of protein evolution have changed across the tree of life requires a mathematical framework. Protein evolution can be modeled in many different ways, but the simplest framework involves the use of a single 20 × 20 matrix of instantaneous rates of amino acid substitution to calculate the probabilities of the site patterns in a multiple sequence alignment (MSA) [10]. Pandey and Braun [7] suggested that one could gain insights into the ways that protein evolution has changed over evolutionary time by comparing rate matrices for different groups of organisms. More precisely, Pandey and Braun [7] suggested that comparing of rate matrices estimated using the 20-state GTR (general time-reversible) model (i.e., the nucleotide GTR model [11,12] extended to the amino acid alphabet) would be useful. Restricting consideration to the GTR_20_ model is desirable because it can yield a single value that characterizes the relative exchangeability (RE) for each pair of amino acids; in contrast, the simplest unrestricted model of amino acid evolution [9] yields two values (i.e., one for amino acid *X*→amino acid *Y* substitutions and a second for *Y*→*X* substitutions). The GTR_20_ model also has fewer free parameters than the simplest unrestricted model (GTR_20_ has 208 free parameters whereas the unrestricted model has 398 free parameters). GTR model parameters are typically written as a diagonal matrix (**Π**) of equilibrium state frequencies and a symmetric matrix of relative exchangeability (RE) parameters [13], often called the ***R*** matrix. Each of the REs in the ***R*** matrix is a single value that can be used to characterize the patterns of sequence evolution for each pair of amino acids.

REs reflect the rates at which non-synonymous mutations enter populations and the probability that those novel non-synonymous changes become fixed as substitutions. Thus, REs reflect processes at the molecular and cellular levels (e.g., the mutational rate and spectrum, the structure of the genetic code, and the impact of amino acid changes on protein structure and function) and population-level processes (e.g., the relative impacts of selection and drift on non-synonymous polymorphisms). Despite the complexity of RE values, the relationship between REs and differences in the physicochemical properties of amino acids has long been clear (see Braun [14] for review). In general, pairs of similar amino acids (conservative substitutions) have large REs, whereas pairs of dissimilar amino acids (radical substitutions) have small REs. Calculations based on REs estimated using MSAs for unrelated proteins from a specific clade can be used to identify the clade of origin for protein MSAs that were not used to estimate the RE values [7,8,9]. This suggests that amino acid substitutions may be more conservative or less conservative depending on the clade under consideration. In principle, it is straightforward to examine differences among clades in their REs (and equilibrium amino acid frequencies): estimate GTR_20_ model parameters for a set of clades across the tree of life and compare the values of those parameter estimates. We believe that a large-scale effort to leverage the relative ease of obtaining maximum likelihood (ML) estimates of RE values across the tree of life will yield biological insights.

Despite the potential benefits associated with using the GTR_20_ model to estimate REs, this approach presents two fundamental challenges. First, even though the GTR_20_ model has fewer free parameters than other potential models of protein evolution, it is still relatively parameter rich. In principle, individual proteins might be unable to provide enough information to yield accurate estimates of all GTR_20_ model parameters; after all, the GTR_20_ model has 208 free parameters and the MSAs for most individual proteins are only 280–600 amino acids in length [15]. Second, the GTR_20_ model assumes that both the REs and the equilibrium frequencies of amino acids remain constant over evolutionary time (i.e., it assumes time-reversibility). However, there is substantial evidence that patterns of amino acid substitution differ among clades [6,7,8,9], indicating that the time-reversibility assumption cannot hold across the tree of life. However, we do not believe that either of these challenges are insurmountable. 

The simplest way to overcome the challenge presented by the high dimensionality of the GTR_20_ model is to use a large number of proteins to estimate parameters [13,16]. We will refer to the process of parameter estimation as “training” because this approach separates the estimation of ***R*** matrix parameters from the use of that ***R*** matrix in phylogenetic analyses. This approach has been extensively used in phylogenetics; in fact, one of the earliest studies that employed ML to estimate a phylogeny using protein data [17] used REs based on the Dayhoff et al. [18] PAM matrix. The PAM matrix was trained with MSAs for 71 groups of proteins with diverse structures and functions using an approximate method to estimate REs. Subsequent studies used matrices estimated a number of “generalized” models of protein evolution using larger training datasets (e.g., the JTT [19], VT [20], WAG [13], LG [16], and Q.pfam [8] models). Generalized model ***R*** matrices reflect patterns of sequence evolution averaged across proteins and across the tree of life, so they cannot provide information about the differences among clades. Other studies have eschewed this broad sampling of proteins and taxa, focusing on specific taxa and/or proteins, such as the proteins encoded by specific viruses [21,22,23,24] or proteins encoded by organelle genomes [25,26,27,28]. Concerted efforts to estimate REs for diverse proteins from specific groups of free-living taxa (i.e., clade-specific generalized models) have only recently been undertaken [6,7,8].

The time-reversibility assumption is irrelevant if protein sequence evolution is *locally time-reversible*. In other words, time-reversibility may hold (at least approximately) within a specific clade even though it does not hold at the scale of the tree of life. This changes the nature of the challenge posed by the time-reversibility assumption; the assumption is only problematic if violations of time-reversibility within clades are large enough to distort the results of comparisons of the RE values estimated using the GTR_20_ model. Pandey and Braun [7] found: (1) that models fit could be used as a classifier (i.e., the best-fitting model for an MSA was the model trained on the clade of origin >70% of the time) and (2) that cluster analysis using distances between the RE values for different models yields a tree that separates vertebrates from plants and microbes. That study focused on eukaryotic models, so it needs to be extended to bacteria and archaea. Finding evidence that either of the Pandey and Braun [7] results held more broadly across the tree of life would corroborate the hypothesis that violations of local time-reversibility are not especially problematic. Such corroboration might take the form of evidence that REs exhibit a historical signal (i.e., patterns in the data that are congruent with at least some parts of the tree of life), an ecological signal (i.e., strong differences between RE estimates for taxa living in distinct environments, such as thermophiles versus mesophiles), or both signals. Obviously, it is possible for some historical and/or ecological signals to emerge and other signals to be distorted, so it is impossible to rule out the existence of any problems linked to violations of the time-reversibility assumption. However, the existence of either (or both) signals in the RE data would provide evidence that our estimates of those parameters suffice for the goals of this study.

The results of studies focused on comparisons of proteome-wide variation in amino acid frequencies, which has received substantial study, provide a framework that might be useful for understanding comparisons of REs. Those studies have used observed proportions of each amino acid instead of ML estimates of the **Π** matrix parameters. The use of observed amino acid proportions makes it possible to examine proteome-wide amino acid compositions for individual taxa (estimating **Π** matrix parameters by ML requires MSAs), and it is less computational burdensome than the full ML approach. This approach has revealed many correlates of proteome-wide amino acid composition, such as genomic GC content and extremophilic lifestyles [29,30,31,32,33]. This raises the possibility that a similar signal related to genomic GC content and lifestyle will be evident in RE data. Regarding the differences among proteins, one major axis of variation in protein composition appears to differentiate ribosomal proteins from other proteins [34]. For this reason, we have separately estimated models for ribosomal proteins and the broader proteome and compared those models to determine whether the REs differed between these protein types.

To better understand the structure of “protein evolutionary model space”, we generated clade-specific models for multiple lineages of bacteria, archaea, and eukaryotes. We trained these clade-specific models by estimating GTR_20_ model parameters from many arbitrarily selected MSAs of homologous proteins from each taxonomic group. To extend this set of clade-specific generalized models, we also estimated model parameters for clade-specific ribosomal protein models using subsets of a concatenated ribosomal protein MSA generated by Hug et al. [35]. These clade-specific models allowed us to ask several questions. First, what type(s) of signal(s) are present in clade-specific model REs? In principle, there could be a phylogenetic signal, an ecological signal, or some mixture of both types of signals. We addressed this question by examining the structure of a clustering diagram generated using distances among models and then comparing that “tree of models” with estimates of the tree of life to look for informative similarities. Second, do the best models for individual proteins cluster near a taxon-specific mean? To answer this question, we used the approach described by Pandey and Braun [7], who asked whether likelihood scores calculated using a set of clade-specific models could be used to identify the source of MSAs that were not present in training data. Third, which clade-specific model parameters exhibit differences? For this, we examined differences among models at the level of the REs (the ***R*** matrix parameters) and equilibrium amino acid frequencies (the **Π** matrix parameters). We answered these three questions using the clade-specific generalized models and clade-specific ribosomal protein models. This led to a fourth question: do the REs and equilibrium amino acid frequencies for generalized models and ribosomal protein models differ? Finally, we discuss the biological implications of the RE estimates and the potential utility of the clade-specific models estimated as part of this study for future phylogenomic studies. We believe that the results of these analyses provide a picture of the structure of protein evolutionary model space for the tree of life.

## 2. Materials and Methods

### 2.1. Estimating Clade-Specific Models

We selected 19 clades spread across the tree of life, using the estimate of the tree of life from Hug et al. [35] (hereafter called the Hug tree) to guide our taxon selection. Then, we trained generalized clade-specific models using MSAs for homologous proteins from representative members of those clades. To do this, we identified whole genome assemblies for selected taxa with annotated genome assemblies in 19 different clades (5 archaea, 2 eukaryotes, and 12 bacteria). We collected files of annotated protein sequences for 11 to 37 taxa in each clade (see Appendix A for the complete taxon list), giving preference to taxa present in the Hug tree. Then, we clustered and aligned proteins in each clade using usearch [36], using a 50% similarity cutoff to cluster and retaining clusters with at least four sequences without any further modifications. The MSAs generated using usearch were then arbitrarily assigned to training and validation sets. The training sets were used to estimate clade-specific model parameters using the ReplacementMatrix server [37]. The ReplacementMatrix server generates a model of evolution by estimating a tree for each MSA and then finding the globally optimal parameters for the GTR_20_ model with Γ-distributed rates across sites given all the MSAs and trees. The use of different trees for each MSA meant that any gene tree-species tree discordance, which was expected due to processes such as horizontal gene transfer [38], would not bias estimates of the rate matrix parameters. The training sets for all clades except Thermoprotei comprised 1000 MSAs (we obtained fewer clusters from Thermoprotei, so that training set was limited to 640 MSAs). We call the REs (the ***R*** matrix parameters) and equilibrium amino acid frequencies (π_X_ values for each amino acid, i.e., the diagonal **Π** matrix elements) that the ReplacementMatrix server estimated for each clade the “generalized model” for that clade.

We trained the clade-specific ribosomal protein models using MSAs directly extracted from the Hug et al. [35] data matrix, which is a concatenated set of ribosomal protein MSAs. For this analysis, we expanded the number of focal clades to 43. We used the Hug et al. [35] MSAs as published, with the exception of eliminating all gap/missing columns in each of the taxon subsets that we selected. We estimated GTR_20_ parameters using IQ-TREE v.1.6.12 [39], modeling among-site rate heterogeneity using invariant sites plus Γ-distributed rates (i.e., the GTR + I + G4 model) and allowing IQ-TREE to search for the ML tree.

### 2.2. Analyzing Model Space and Assessing the Performance of Model Fit as a Classifier

To examine the structure of model space, we calculated Euclidean distances among REs for each model (Perl scripts that normalize ***R*** matrix parameters to sum to one and calculate distances are available from https://github.com/ebraun68/protmodels; accessed on 20 December 2022). We visualized model space by clustering the Euclidean distances between models with neighbor-joining [40] in PAUP* [41]. We assessed the congruence between the model clustering trees and the Hug tree using matching distances [42,43] calculated using PAUP*. We tested the hypothesis that the structure of model space resembles the tree of life (i.e., the hypothesis that model parameters have a phylogenetic signal) by comparing the matching distance between the Hug tree and the model clustering trees to the matching distances between the Hug tree and random trees (generated by PAUP* using the assumption that trees are equiprobable). This strategy resembles the approach that Penny et al. [44] used to show that the theory of evolution is falsifiable. 

We evaluated the ability of model fit to classify validation set MSAs. We identified the best-fitting model by fitting all models to a single tree for each validation set MSA. The tree for each MSA was obtained by conducting a tree search using the Q.pfam model [8], combined with empirical amino acid frequencies and Γ-distributed rates across sites (i.e., the Q.pfam + F + G4 model). Then, we obtained BIC [45] scores for each model given that tree, using the ***R*** matrix + F + G4 model in all cases (we parsed the BIC because it is the default model selection criterion in IQ-TREE, but model fit rankings would be identical regardless of whether we used BIC, AIC, AIC*_c_*, or likelihood because all models that we examined have the same dimensions). The script for this analysis is available from https://github.com/ebraun68/protmodels (accessed on 20 December 2022). We used recall to examine the performance of models as classifiers; since we conducted separate analyses of each validation set, recall was simply the number of true positives (cases where the MSA had the best fit to the appropriate clade-specific model) divided by the number of MSAs in each validation set. We scored true positives at two taxonomic levels. *Precise true positives* were cases where a validation set MSA from a specific bacterial clade had the model trained on MSAs from that specific clade. *Domain-level true positives* were cases where a bacterial MSA was called a true positive if the best-fitting model was any of the models trained using bacterial MSAs. We repeated the model fit analyses on the training set MSAs to serve as a comparison to the model fit analysis for the validation set.

We examined the fit of two model sets, the novel clade-specific models generated for this study (both the general models and the ribosomal protein models), and a “complete” model set. The complete model set comprised: (1) the novel clade-specific models, (2) generalized models (e.g., the PAM [18], JTT [19], VT [20], WAG [13], LG [16], and Q.pfam [8] models), and (3) published clade-specific models [7,8,46]. All models are available from https://github.com/ebraun68/protmodels (accessed on 20 December 2022) as PAML format rate matrices. The model files, the MSAs used for training data, and the validation set MSAs are available from Zenodo [47].

## 3. Results and Discussion

### 3.1. The Structure of Protein Model Space Resembles the Tree of Life

The trees of models (i.e., the clustering diagrams based on Euclidean distances among models; Figure 1a,b) showed clear similarities to the tree of life (Figure 1c). The topological distances between the Hug tree and the clustering diagrams based on generalized models and ribosomal protein models were smaller than expected for distances between random trees of with the same number of taxa (Figure 1d,e). The most important similarity between the clustering diagrams for models and the tree of life is the fact that most or all bacterial models were separated from the archaeal and eukaryotic models. The existence of a bipartition separating bacteria from archaea and eukaryotes is long-standing [48] and, at this point, uncontroversial (reviewed by Eme et al. [49]). The only exception to this division was observed in the clustering diagram for general models: Methanomicrobia (archaea) and Selanomonadales (bacteria) formed a cluster separate from either group (Figure 1a). In contrast, the bipartition separating bacteria from archaea + eukaryotes was perfect when we clustered ribosomal protein models (Figure 1b), although the position of the Methanomicrobia ribosomal protein model differed from the current best estimate of archaeal phylogeny (Figure 1c), which places the DPANN group sister to the other archaea (see also Castelle et al. [50] and Williams et al. [51]). The tree based on clustering ribosomal protein models exhibited additional similarities to the tree of life (these similarities involved the taxa presented using bold text in Figure 1b,c). Specifically, the clustering diagram for clade-specific ribosomal protein models and the Hug tree both included: (1) the bipartition separating eukaryotes, Thaumarchaeota, and Thermoprotei from other taxa (also see Williams et al. [52]); (2) the subtree comprising Rhodospirillales, Rhodobacteriacae, and Rhizobiales; and (3) the subtree comprising Levybacteria, Gottesmanbacteria, and Roizmannbacteria within the “candidate phyla radiation” (CPR [53,54]). These results suggest that there is a phylogenetic signal in protein models, especially the differences between bacteria and archaea + eukaryotes.

We refer to the clustering diagrams for models of protein evolution (Figure 1a,b) as “trees of models” to emphasize that they are not estimates of phylogeny. The trees of models generated by clustering distances among models are only expected to exhibit congruence with phylogeny if three conditions are met. First, the estimates of model parameters must not be strongly biased due to violations of the GTR_20_ model; this can be viewed as a restatement of our local time-reversibility assumption. Second, the patterns of protein evolution captured by the models must change over an appropriate timescale and exhibit limited convergence. If proteome-wide REs changed very rapidly (relative to the timescales for the evolutionary history of clades we chose), we are unlikely to recover any historical signals at the scale of the tree of life. Likewise, proteome-wide convergence in REs could have an impact on the topology of the tree of models; in fact, we expected strong ecological signals in the REs to yield a tree that included bipartitions that divided the taxa into subsets with similar lifestyles (e.g., a thermophile versus non-thermophile bipartition). Finally, the estimates of distances among models (in this case, Euclidean distances among the normalized vectors of RE values) must yield useful estimates of actual distances between models. These factors mean that it would have been appropriate to interpret failure to find congruence between trees of models and the Hug tree with caution; failure to find evidence of congruence could have meant that there was no historical signal or it could have been evidence for analytical biases that distorted the trees of the models. On the other hand, evidence of congruence is likely to be meaningful. 

The topological congruence between the trees of models and the tree of life (Figure 1) provides evidence that there is a historical signal in the RE values. Indeed, these results are consistent with a single large-scale model shift on the branch separating bacteria from archaea and eukaryotes combined with a modest historical signal elsewhere in the tree. Of course, it remains possible that using other methods to extract information about changes in the patterns of protein sequence evolution might reveal evidence for an even stronger historical signal. However, these analyses make it clear that there is at least some historical signal in the RE data—the patterns of protein sequence evolution—that can be extracted using our methodology.

### 3.2. Models of Protein Sequence Evolution Can Sometimes Be Used as Domain-Level Classifiers

A straightforward way to examine the ability of our clade-specific models (sets of REs estimated using each training set) to predict patterns of sequence evolution was to examine recall (true positives divided by the total number of MSAs from each source) when the models were used to classify validation set MSAs. Pandey and Braun [7] reported that recall is typically >70% when clade-specific models are used to classify MSAs comprising sequences from specific eukaryotic clades (birds, mammals, plants, yeasts, and oomycetes). In fact, Pandey and Braun [7] found that recall was >70% in all cases if one viewed the cases where the best-fitting model for avian MSAs and mammalian MSAs were trained using bird or mammal data (i.e., when we viewed the classification of avian and mammalian MSAs as vertebrate MSAs as true positives). The recall was only >70% for some of the models estimated as part of this work, and in those cases, the >70% recall was only achieved when the domain-level true positives were considered (i.e., when the classification of the MSAs as bacterial, archaeal, or eukaryotic was viewed as the endpoints; see bold values in the last three columns of Table 1; see also Appendix A). Moreover, even when we limited consideration to domain-level classification, there were eight (out of 19) clades where recall was <70%, emphasizing the limited utility of model fit as a classifier for MSAs from these taxa. We observed virtually identical results when we classified individual training set MSAs (Appendix A). Recall was much lower when we limited our consideration to precise true positives (i.e., when we attempted to classify MSAs at the level of the specific clades used for model training). Our inability to use model fit as a precise classifier was further emphasized by the observation that Thermoprotei and Halobacteriaceae were the only clades with >50% precise matches (Table 1). The high precise match percentages in those two cases probably reflected the distinctive nature of the selective pressures on those archaeal lineages (e.g., the high intracellular ion concentrations of Halobacteriaceae [55]). Thus, we interpret the distinctive nature of the models for those lineages to be evidence for an ecological signal.

Since Thermoprotei and Halobacteriaceae had large percentages of precise matches, we examined differences between the REs for each of those taxa and the average REs for other archaea. The specific REs with the largest differences appeared to be related to the equilibrium amino acid frequencies for each model, although the nature of this relationship was complex. The most extreme cases for Thermoprotei included R–K, S–T, and N–G exchanges (elevated in Thermoprotei) and Q–K, R–Q, and C–S exchanges (reduced in Thermoprotei; Appendix A). Relative to the other archaea, the equilibrium frequency for arginine was elevated in Thermoprotei (π_R_ range for archaea = 0.0342–0.0771; Thermoprotei π_R_ was the maximum; Appendix A), whereas serine and threonine were both reduced in Thermoprotei (π_S_ range = 0.0568–0.0794 and π_T_ range = 0.0407–0.0651; Thermoprotei π_S_ and π_T_ were the minimum values; Appendix A). It is possible that these estimated equilibrium frequencies for Thermoprotei were related to the generally elevated arginine frequency and reduced serine frequency in thermostable proteins [56], although we note that we examined other thermophilic lineages (Thaumarchaeota and DPANN also include thermophiles). In Halobacteriaceae, the elevated REs included I–V, H–Y, and Q–K and the reduced REs included P–Y, L–V, and I–Y (Appendix A); the estimated equilibrium frequencies for isoleucine, histidine, lysine, and tyrosine for Halobacteriaceae were all the minimal values for the archaea we examined (Appendix A). A prominent feature of Halobacteriaceae proteins relative to other taxa is a much higher ratio of acidic to basic residues, which is known to improve protein solubility given high salt concentrations [57]. This difference was evident in our estimated equilibrium frequencies (the acidic to basic residue ratio based on equilibrium amino acid frequencies was 2.43 for Halobacteriaceae, but it ranged from 0.792 to 1.06 for the other archaea we examined; Appendix A). However, only one of the outlier REs (if we use top three and bottom three RE differences to define outlier) involved a basic or acidic residue, suggesting that the shifts in those amino acid frequencies did not have a major impact on the REs.

The bacterial clade with the highest precise match percentage was Micrococcineae, a kind of actinobacteria (which are high-GC Gram-positive bacteria). However, the precise true positive percentage for Micrococcineae was only 36.3% (Table 1). Some validation sets were found to have very low percentages of precise true positives, with the most extreme being 1.5% (for Oceanospirillales, a clade within γ-proteobacteria, a Gram-negative lineage). We note, however, that having low proportions of MSAs with precise matches is not a feature of proteins from γ-proteobacteria; the precise match percentages for the other γ-proteobacteria that we examined (Alteromonadaceae, Chromatiales, Enterobacteriaceae, and Xanthomonadaceae) ranged from 7.3% to 21.1%. Regardless of the specific details, it seems likely that the low recall when model fit was used as a classifier (at least when we used precise matches to examine recall) indicates that there is a high degree of variation among individual proteins in their patterns of sequence evolution.

Many different protein models trained on a variety of datasets have been published; most of these models (e.g., the WAG [13] and LG [16] models) are similar to our general models in that they were trained on a wide variety of proteins. Most published models were trained using MSAs that include a wide variety of organisms (i.e., the models are not clade-specific). Including those published models in our model fit analyses further emphasized the limited ability to classify protein MSAs using model fit (Appendix A). As with the analyses using new models, this is likely to reflect variation among proteins in their patterns of evolution. The percentage of cases where our new models (either general or ribosomal protein) had the best fit to a protein MSA ranged from 85.16% (for Thermoprotei) to 23% (for Coelomata), with a median of 44.38%. The percentage of times that a published generalized model (i.e., models trained on proteins from diverse taxa, such as Q.pfam/Q.pfam-gb [8], WAG [13], LG [16], JTT [19], and VT [20]) had the best fit ranged from a minimum of 5.78% (Thermoprotei) to a maximum of 53.38% (Coelomate animals). The percentage of times that a published clade-specific model (i.e., the clade-specific models from Pandey and Braun [7] and Minh et al. [8]) had the best fit ranged from a minimum of 9.06% (Thermoprotei) to a maximum of 50.51% (Pezizomycotina). In general, clades with a high percentage of matches to the new models also had a high recall at the domain level (compare Table 1 and Appendix A). These results further emphasize the high degree of variation among individual proteins in their patterns of sequence evolution.

### 3.3. Genomic Base Composition Affects Amino Acid Frequencies and Relative Exchangeabilities

The bacterial and archaeal clades with relatively high recall (Halobacteriaceae and Micrococcineae) were also found to have the highest genomic GC content (Table 1). This led us to speculate that models of protein evolution trained using MSAs from GC-rich taxa might be more successful as classifiers than those trained using GC-poor taxa. If true, this suggests that the patterns of protein evolution for GC-rich taxa, which are what we captured in our models, might be especially distinctive relative to GC-poor taxa. The hypothesis that models for protein evolution in GC-rich taxa are especially distinctive, both relative to each other and relative to the models for GC-poor taxa, is consistent with the observation that the other archaeal group with a very high precise match percentage (Thermoprotei) had the second highest genomic GC content within archaea (Table 1). This led us to ask whether there was a correlation between GC content and the precise match percentage; however, we did not find a correlation (Spearman’s ρ = 0.32383, *p* = 0.17622). Thus, it seems likely that the observation that the most GC-rich clades in our sample also had relatively high precise match percentages was coincidental.

Some REs did appear to be related to genomic GC content despite the absence of a correlation between the precise match percentage and GC content. Focusing on bacteria, where we sampled the largest number of clades, the mean REs for high-GC bacteria (54.4% to 69.7% GC) and those for low-GC bacteria (37.7% to 52.2% GC) differed in specific ways: (1) the three REs most elevated in high-GC taxa all involved aromatic residues (F–Y, W–Y, and H–Y), and (2) the three REs most elevated in high-GC taxa all involved small residues (A–S, A–C, and C–S) (see Appendix A for all RE comparisons). The ML estimates of equilibrium amino acid frequencies also exhibited a strong relationship with genomic GC content (Figure 2 and Appendix A). The second result is consistent with earlier work that showed that GC content has a proteome-wide impact on observed amino acid composition [29,58]. The correlation between amino acid composition and genomic GC content is likely to have an impact on estimates of REs; after all, the **Π** matrix (equilibrium amino acid frequencies) and ***R*** matrix (REs) are interrelated. Phenylalanine and tyrosine are both encoded by GC-poor codons and alanine is encoded by a GC-rich codon; this may explain some of the RE differences between the models for GC-rich and GC-poor taxa. However, the equilibrium frequency of tryptophan (π_W_) was only weakly correlated with genomic GC content (Appendix A), and the other amino acids involved in REs with the largest differences between GC-rich and GC-poor taxa were not correlated with genomic GC content. Overall, these results provide evidence that genomic GC content has a major impact on amino acid composition and a modest impact on REs. However, genomic GC content did not affect the overall model in such a way the fit became universally better as a classifier for either the high-GC or low-GC subsets of taxa.

### 3.4. Generalized Models and Ribosomal Protein Models Exhibit Specific Differences

The RE values that exhibited the largest differences when general models and ribosomal protein models were compared fit expectations of the Pandey and Braun [7] “rule of opposites”. The rule of opposites is the observation that the REs for pairs of amino acids that have low frequencies in specific structural environments are elevated; the name of the rule reflects the observation that the behavior of REs is the opposite of equilibrium frequencies. Pandey and Braun [7] explored the relationship between REs and solvent exposure in globular proteins; they found that REs for pairs of polar amino acids are elevated for buried residues and that REs for pairs of hydrophobic amino acids are elevated for solvent-exposed residues. Gordon et al. [28] observed a similar pattern in their comparison of transmembrane helices and extramembrane residues. Although this study did not consider protein structure, the observation that the RE value in ribosomal protein models with the greatest elevation relative to the general models involved the acidic amino acids (D–E; Figure 3) is a similar result because acidic residues are rare in ribosomal proteins [59]. Likewise, the RE that was most elevated in general models relative to ribosomal protein models involved basic amino acids (R–K), which are common in ribosomal proteins. A comparison of high- and low-GC bacteria highlighted F–Y as the most elevated RE in high-GC bacteria, and the estimated equilibrium frequencies of both phenylalanine and tyrosine were relatively low in high-GC taxa (Appendix A). Thus, comparisons between generalized models and ribosomal protein models and (at least to some degree) between models for high- and low-GC bacteria indicate that the rule of opposites extends beyond different structural environments to complete MSAs.

The rule of opposites is likely to reflect, at least in part, our assumption that models of protein evolution are locally time reversible. After all, large REs are necessary to explain substitutions that result in low-frequency amino acids if time-reversibility holds. This means that the RE values for rare–rare pairs must be elevated if the MSAs provide evidence that at least some substitutions involving pairs of rare amino acids occurred. However, we believe that finding RE differences that are consistent with the rule in multiple matrices provides evidence that the rule is likely to have a biological basis (as opposed to a purely methodological basis). This can be illustrated by considering two alternative hypotheses regarding substitutions involving rare amino acids: (1) when present, rare amino acids are typically necessary for specific functions and therefore highly conserved, and (2) rare amino acids are not especially conserved. Hypothesis 1 predicts a very low number of rare–rare exchanges and will therefore result in low RE estimates for those pairs if the dataset used to estimate the ***R*** matrix is large enough. In contrast, hypothesis 2 predicts that the number of rare–rare exchanges relative to their frequencies will be large, so it predicts high RE estimates. Of course, both hypotheses predict that there will be a small number of site patterns that can provide information about rare–rare REs, which will increase the variance of the rare–rare RE estimates. Thus, one could reconcile hypothesis 1 with the observation that some rare–rare RE estimates are high in specific ***R*** matrices by invoking the high variance expected for those RE estimates. On the other hand, hypothesis 1 cannot be reconciled with the observation that rare–rare RE estimates are high across multiple ***R*** matrices estimated using independent datasets; that is only expected if hypothesis 2 is correct. Both Pandey and Braun [7] and Gordon et al. [28] examined multiple ***R*** matrices trained using multiple independent datasets, and they found high rare–rare RE estimates for distinct structural environments. The observation that rare–rare exchanges are elevated in the ribosomal protein versus generalized model comparisons and the results of the comparisons between models for GC-rich and GC-poor taxa (see Section 3.3) further corroborate the second hypothesis.

### 3.5. Aromatic–Aromatic REs Differentiate Bacterial versus Archaeal/Eukaryotic Models

Most RE values in the average bacterial model and the average archaeal/eukaryotic model were quite similar (note the large number of RE differences that are near zero in Figure 3). A similar pattern emerged when we examined a bivariate plot of bacterial versus archaeal/eukaryotic RE differences for general and ribosomal protein models (Figure 4) The F–Y exchange was the most extreme difference between bacteria and archaea + eukaryotes; it was elevated in the average bacterial model of both types relative to the average archaeal/eukaryotic model (Figure 4).

Other RE differences were less extreme, but a number were still outliers. For example, the histidine–tyrosine (H–Y; Figure 4) REs were elevated in the average general archaeal/eukaryotic model (relative to the average general bacterial model), but the opposite was true for ribosomal protein models. In both cases, these exchanges had extreme ranks (this ranking was similar to the rankings shown in Figure 3, with a rank of 1 indicating the most elevated RE in the average archaeal/eukaryotic model and a rank of 190 indicating the most elevated RE in the average bacterial model). Specifically, the H–Y RE had a rank of 1 for general models and a rank of 184 for ribosomal protein models (in contrast to F–Y, which had a rank of 190 in both general models). F–W and W–Y were elevated in the average archaeal/eukaryotic ribosomal protein models (ranks of 1 and 2, respectively), but they were less different from the majority of REs for general models (F–W and W–Y had ranks of 178 and 184, respectively, for general models). Although some outlier RE differences did not involve aromatic residues (including the very conservative I–V exchange; Figure 4), we emphasize that four of the six possible aromatic–aromatic exchanges were outliers.

### 3.6. Models Trained Using Methanomicrobia MSAs Are Outliers within Archaea

The tree of generalized models (Figure 1a) and the tree of ribosomal protein models (Figure 1b) both included a bipartition that divided Methanomicrobia + bacteria from the other archaea and eukaryotes. In fact, the generalized model for Methanomicrobia clustered to the generalized model for a bacterial clade (Selenomonadales). This suggests that, based on REs, Methanomicrobia had the most distinctive patterns of protein sequence evolution relative to other archaea. The largest RE difference between Methanomicrobia and the remaining archaea and eukaryotes was the F–Y pair (Appendix A), with the Methanomicrobia model having a larger value than the mean for the other archaea and eukaryotes. Several amino acid pairs that include cysteine also had larger REs in the generalized Methanomicrobia model than in the other generalized archaea + eukaryotes models; more specifically, C–S, C–Y, and C–F had ranks of two, three, and four, respectively, when the differences between the Methanomicrobia REs were subtracted from the mean normalized REs for other archaea + eukaryotes. The C–W difference was ranked eighth in the same analysis (Appendix A). Since three of these examples reflected cysteine–aromatic pairs, the generally high variation in the REs for aromatic residues could have contributed to the variation in those pairs. However, cysteine has an unusual role in methanogen proteomes; methanogens have cysteine-rich proteomes, and they sometimes have an unusual cysteinyl–tRNA synthesis pathway [60]. Our estimate of the equilibrium frequency of cysteine in the Methanomicrobia model (π_C_ = 0.00953) was the highest value out of all the archaeal models (π_C_ range for archaea = 0.00325–0.00953; median π_C_ = 0.0078; Appendix A). The observation that the high REs involving cysteine occurred in an archaeote with an elevated (rather than reduced) cysteine frequency was not consistent with the rule of opposites, but we nevertheless view the result as likely to be biologically relevant given the role of cysteine in methanogen proteomes.

The unexpected placement of Methanomicrobia in the trees of models might lead one to hypothesize that those models were either especially unique, suggesting a high percentage of precise matches for Methanomicrobia, or that the REs for Methanomicrobia were intermediate between those for archaea and bacteria, suggesting that there would be many cases where a bacterial model had the best fit to an MSA from Methanomicrobia. Neither of these predictions were true (Table 1); the percentage of precise matches was relatively low, and the percentage of cases where a bacterial model had the best fit to an MSA from Methanomicrobia was not especially high (it was actually lower than the percentages for two other archaeal lineages, Thaumarchaeota and DPANN; Table 1). Taken as a whole, these results suggest patterns of sequence evolution for Methanomicrobia have some unusual features, but they also indicate that patterns of sequence evolution in that clade are not exceptionally distinctive relative to the other lineages that we examined (at least using the metrics employed in this study).

### 3.7. Does Our Approach Provide an Accurate Picture of Protein Evolutionary Model Space?

Unlike many descriptive statistics used in comparative genomics, REs must be estimated from a set of protein MSAs using a specific model. This raises a question: is the model we chose appropriate for our goals? Our primary goals were to ask: (1) Is there evidence for a signal (or signals) in the clade-specific RE estimates? and (2) do the best models for individual proteins cluster near clade-specific means? To answer those questions, we estimated clade-specific REs using the GTR_20_ + Γ model, a general approach that can be traced to the pioneering work of Dayhoff et al. [18]. However, as Thorne et al. [61] stated, the “…problem with the Dayhoff approach is that it effectively models the replacement process at the ‘average’ site in the ‘average’ protein”. Thus, our approach ignored variation among sites within proteins and among proteins.

The practice of modeling protein evolution using a single rate matrix has been criticized precisely because patterns of evolution are known to vary among sites within proteins (see Goldstein and Pollock [62] for review). Many lines of evidence corroborate the hypothesis that different sites in functional proteins can only be occupied by specific sets of amino acids rather than the complete set of 20 amino acids. Crooks and Brenner [63] argued that the existence of among-site variation in amino acid propensities explains the observation that database searches using profiles identify more distant homologs than searches using a single query. Deep mutagenesis studies have revealed that the sets of amino acids permitted at specific sites resemble profiles obtained by comparing homologous proteins (e.g., Melamed et al. [64]), although it is important to note that those sets of amino acids tend to have similar physicochemical properties. Single rate matrix models violate the assumption of site-specific amino acid preferences because they predict that the probability that, given sufficient evolutionary time, any specific amino acid *X* will be aligned with a different amino acid *Y* is given by π_X_π_Y_. This implies that all 20 amino acids are permitted at all sites (although it is important to note that this disturbing behavior appears in the limit of large evolutionary divergences). Two basic approaches have been used to model site-specific amino acid preferences in phylogenetic analyses: (1) incorporating protein structures into models of protein evolution [7,46,61,65,66,67,68] and (2) using “profiles” with different amino acid frequencies [69,70,71]. Both of those approaches are typically implemented as mixture models. Could those models provide a better picture of evolutionary model space for proteins?

Pandey and Braun [7] already used a simple mixture model in a similar analysis and did not find that it had benefits for understanding model space; trees of models based on Euclidean distances among models were quite similar for solvent-exposed and buried sites in globular proteins. Although it is possible that more sophisticated ways of incorporating structure might yield new insights, there is no clear evidence that straightforward extensions to existing methods to incorporate protein structure are especially beneficial. Methods based on site-specific profiles [69,70,71] have been used in many phylogenetic studies (e.g., Williams et al. [52]), sometimes in combination with the subdivision of protein MSAs using structure (e.g., Pandey and Braun [46,68]). However, there are no obvious ways to compare profile models trained using data from different clades. Moreover, the disturbing behavior of simple rate matrix models (i.e., the fact that all sites can accept all 20 amino acids) only appears at large evolutionary divergences; at lower divergences, they yield site patterns dominated by physiochemically similar amino acids, similar to models that incorporate site-specific amino acid propensities. Thus, we believe that it is reasonable to view simple rate matrices as approximating models that are valid when evolutionary divergence is limited that also have the benefit of simplifying parameter estimation and comparison.

The other concern associated with the use of generalized rate matrix models is the fact that the best model of evolution for individual proteins may differ from the average pattern in the rate matrix. There is ample evidence for variation among individual proteins. Braun [14] proposed a model framework that adjusts the REs of existing models using the physicochemical properties of amino acids and found variation among proteins (this model framework limits the number of free parameters that need to be optimized). Del Amparo and Arenas [72] further developed the idea of looking at the ways that the model differs among proteins by optimizing all rate matrix parameters for two individual proteins (HIV protease and integrase); that approach yielded models with better fits to other protease and integrase MSAs than any other available models (even models trained using other HIV proteins). However, the existence of model variation among proteins invalidates the possibility that the best models for individual proteins in a clade tend to be close to clade-specific averages. Our clade-specific generalized models are estimates of those averages. In principle, the best models for individual proteins might tend to cluster very close to a clade-specific generalized model or they might be dispersed in model space (Figure 5). Work using models trained on MSAs from several eukaryotic clades indicates that model fit is a fairly good classifier [7,8,9], suggesting the strong clustering of individual protein models (Figure 5a). However, similar analyses in this study (Table 1) indicated that model fit was a poor classifier, suggesting much weaker clustering (Figure 5b).

### 3.8. What Is the Biological Basis for the Structure of Model Space That We Observed?

The observation that model fit had a very limited ability to classify MSAs in this study (Table 1) appears to reflect a high degree of variance among proteins in their best model (e.g., Figure 5b). The fact that model fit was a relatively good classifier in other studies that used similar methods [7,8,9] raises a question: why did this study have different results? One reason could be the very large effective population sizes of prokaryotes, which makes natural selection very efficient [73]. It is reasonable to hypothesize that the unique features of specific proteins might push the patterns of evolution for individual proteins away from each other when selection is very efficient; in contrast, REs for individual taxa might move toward each other (and toward clade-specific means) when mutation and drift dominate. Cases where we found that model fit was a good classifier could reflect clades with similar patterns of selection on most proteins or clades where mutation and drift dominate. Similar patterns of selection are likely to be the case for Halobacteriaceae; high intracellular ion concentrations [55] result in distinctive selection pressures. Whether the same is true for Thermoprotei is less clear, but it warrants further investigation. One potential problem with our effective population size hypothesis is the fact that models for ascomycete yeasts (Saccharomycetes) exhibit high recall [7,8,9] but filamentous ascomycete fungi (Pezizomycotina) models appeared to be poor classifiers in this study; the idea that differences in typical effective population sizes for those clades could result in the observed differences is surprising. Regardless, there is a reasonable biological basis for the patterns we observed in at least one case (Halobacteriaceae) and possible reasons that are related to the strength of selection in other cases.

The REs that appeared to exhibit the largest degree of variation among models also exhibited a specific pattern; the most variable REs consistently involved specific types of amino acids, most notably aromatic residues (Figure 4). This observation has several possible interpretations, the simplest of which could be related to the relatively distinctive chemical properties of aromatic amino acids. Specifically, it has long been recognized that aromatic residues play an important role in protein structure, both via aromatic–aromatic interactions [74,75,76,77] and through interactions between the aromatic rings and nitrogen and sulfur atoms in other amino acids [78,79]. This might make the REs for those amino acid pairs especially sensitive to global shifts in the patterns of selection on the proteome. Another possibility is that there was a major, ancient shift in REs on the branch separating bacteria from archaea + eukaryotes, followed by changes in response to specific factors such as the shifts in genomic base composition or environmental shifts (e.g., high salt concentrations in Halobacteriaceae). Notably, the aromatic amino acids, as well as some of the other amino acids involved in highly variable pairs, are thought to be late additions to the genetic code and to have increased in frequency since the time of the last universal common ancestor of life [80,81,82]. If the late addition of these amino acids is related to variations in REs, then the details of the models of protein evolution might themselves provide a record of evolution.

More speculatively, the hypothesis that there was a major, ancient shift in REs on the branch separating bacteria from archaea + eukaryotes might reflect the “crystallization” of the translation apparatus that was occurring at that time. Woese [83] argued that ancestral cellular systems were so loosely organized that the horizontal transfer of genes encoding proteins involving virtually all of those systems could occur, potentially leading to the displacement of those genes by non-orthologous genes. Those cellular processes would then become more refractory to horizontal transfer over time in a process analogous to crystallization (although it is important to recognize that horizontal gene transfer has remained an important evolutionary process for microbial life even after crystallization [38]). The translation apparatus (i.e., the ribosome and some elongation factors) was established in the last universal common ancestor [83,84]. However, the ribosome continued to change after the divergence of bacteria and archaea [84,85], so much so that Woese [83] believed that the “canonical pattern” for translation proteins was the case where “…bacterial and archaeal versions of [a translation] protein are remarkably different from one another…”. There was an especially provocative similarity between the putative historical signal in the evolutionary model space (Figure 1a,b) and the potential historical signal in the geometry of the ribosome exit tunnel (the ribosome exit tunnel), which has an important role in the folding of nascent proteins [86]). Bowman et al. [84] hypothesized that typical patterns of protein folding coevolved with the exit tunnel and that the clustering of distances between different exit tunnel geometries yields a “tree of exit tunnel geometries” with a topology similar to the tree of life [87]. This makes it tempting to speculate that the REs, which capture the effects of selection and mutational input, are revealing a signal related to selection imposed by the history of the ribosome itself. Unfortunately, the nature of any testable hypotheses that might falsify or corroborate this idea is unclear. Regardless of the basis for the RE shifts, however, the shifts represent an important feature of protein evolution.

### 3.9. Can Our Clade-Specific Models Improve Estimates of the Tree of Life?

The goal of this study was to improve our understanding of the structure of protein evolutionary model space at the scale of the tree of life; we do not believe that our tree of models is an improved estimate of the tree of life itself. However, our results do provide information and resources that have the potential to improve phylogenetic analyses of prokaryotes, including analyses focused on the deepest branches in the tree of life. A major challenge for phylogenetic estimation at the scale of the tree of life is the fact that many individual gene trees differ from the species tree. In fact, some researchers have suggested that the horizontal transfer of genes is common enough that it is only meaningful to discuss the “network of life” rather than the tree of life [38]. The net-like nature of the tree of life raises an important question: are estimates of the trees of life generated using concatenated MSAs, such as the Hug tree, sufficiently accurate? Methods that directly incorporate variation among gene trees do exist; specifically, there are supertree methods that combine phylogenetic trees to produce a more inclusive phylogeny [88], and those methods have the potential to address incongruence among gene trees in an explicit manner. Individual gene trees are expected to exhibit incongruence with the species tree due to genuine discordance [38,89] and estimation errors [90,91], but combining a large set of gene trees to yield a single tree might overcome both sources of error. The supertrees of life has been used in a number of recent studies focused on deep branches in the tree of life (e.g., work by Williams and colleagues [51,52] and the recent, very large-scale tree by Zhu et al. [92]). The Zhu supertree is especially interesting because it is both a very large scale analysis (>10,000 taxa) and it uses a supertree method that is likely to be robust to many different sources of incongruence among gene trees [93]. One thing that is clear is that supertrees require many gene trees as input, so any method with the potential to improve rapid gene tree estimation should improve supertree construction. The availability of many protein models trained using prokaryotic data (such as the clade-specific models we generated) has the potential to do just that and, therefore, improve the construction of an accurate estimate of the (super)tree of life.

Obviously, any set of novel protein models also has the potential to be useful in studies with the goal of predicting protein function using trees (i.e., phylogenomics as it was originally defined [94,95]). However, we also want to express caution regarding the use of single rate matrix models; Spielman [96] recently questioned the necessity of using the best-fitting protein models in phylogenetics. That conclusion was based on a simulation study that generated MSAs using a codon-based model with site-specific amino acid profiles and then analyzed those MSAs using a set of “Dayhoff-type” single rate matrix models, as well as the GTR_20_ model. Those analyses revealed that all models resulted in trees with similar error rates. This is an intriguing result, but it is unclear how it should be viewed in practice. After all, it is necessary to choose a model to estimate a tree, and there is no obvious alternative to model fit that could guide that choice. Perhaps the biggest lesson from Spielman [96] is that it might be more fruitful to determine whether more “biologically realistic” (and potentially much more parameter-rich) models improve single-protein phylogenetic analyses and, if they do, to focus on those models rather than expanding the set of available rate matrices.

The broader goal of developing more realistic protein models raises an important question: what sort of model improvements should the community focus on? Potential areas include: (1) accommodating “net-like” aspects of the tree of life, (2) modeling the variation among proteins and among sites within proteins, (3) relaxing the time-reversibility assumption, and (4) improving comparisons among models. Obviously, there is room to work on all these areas, so we will focus on the issue of model shifts, which is central to this study. First, existing models can accommodate discordance among gene trees; in fact, the method we used to estimate REs [37] optimized parameters on a set of gene trees with the potential to differ. Gene tree-species tree discordance is also unlikely to bias the comparison between our tree of models and the tree of life; the topology for the subset of the Hug tree that we used was very similar to the topology of the Zhu et al. [92] supertree (Appendix A). Moreover, one of the most prominent divisions in our tree of models is the bipartition separating bacteria from archaea + eukaryotes, and that bipartition is hardly controversial.

Shifting from the structure of the tree of life (and the associated network of gene trees) back to the process of amino acid substitution, we acknowledge that there is room to improve the modeling of variation among proteins and among sites within proteins. However, as we argued above (Section 3.7), improving models in this way is unlikely to have an impact on the type of comparisons we conducted. With that said, there have been assertions that the better modeling of variation among sites might be valuable for tree estimation (e.g., by reducing the impact of long-branch attraction [97]), so it is likely to be useful for tree estimation. Simple methods to relax time-reversibility (e.g., using the unrestricted 20-state model [9]) are also unlikely to improve our analyses. Comparing the proportions of cases where the best-fit models were precise matches in studies using time-reversible models (Figure 2 in Minh et al. [8] and Table 3 in Pandey and Braun [7]) to the results of similar analyses in a study using the unrestricted model (Figure 2 of Dang et al. [9]; note that the relevant data in that study were the sums of the best-fitting reversible and non-reversible clade-specific models) revealed similar performance. The unrestricted model does have substantial utility for rooting trees, although we note that there have likely been many model shifts across the tree of life (minimally, this study provides evidence for RE shifts at the base of the tree and at the base of several specific clades, such as Halobacteriaceae). A more principled alternative to calculating Euclidean distances among vectors of REs would be to directly incorporate a “model of model change” into analyses. Recently, Berv et al. [98] used such a method to examine patterns of nucleotide sequence evolution in birds, finding evidence for complex shifts in the patterns of avian genome evolution in the wake of the Chicxulub bolide impact at the end of the Cretaceous period. It would be interesting to see if such an approach could be applied to the much more ancient divergences at the base of the tree of life. Nevertheless, despite the simplicity of the approach we used here, it was able to provide evidence for a phylogenetic signal in our models of protein evolution.

## 4. Conclusions

We hypothesized that comparisons of estimated amino acid REs for various clades, obtained using the GTR_20_ model, would reveal a picture of evolutionary model space for proteins. The trees of models generated by clustering distances among the REs resembled the tree of life, regardless of whether the REs were estimated using arbitrarily selected proteins or ribosomal proteins. There were no obvious clusters that separated taxa that grow in distinct environments or metabolic features, suggesting that phylogeny has a stronger influence on the structure of protein evolutionary model space than ecology. However, the performance of model fit as a classifier did provide evidence that the models for two archaeal clades (Halobacteriaceae and Thermoprotei) were especially distinctive, which could reflect an ecological signal. However, the performance of model fit as a classifier was generally poor; indeed, classification with high recall at the level domain was only possible for a subset of clades. This differs from the results of earlier studies that estimated models for several different eukaryotic groups [7,8,9]. The differences between this study and previous studies in the relative performance of model fit as a classifier probably reflect the differences in the amount of variation among individual proteins in their patterns of sequence evolution. We found that a specific subset of REs tended to exhibit differences among taxa, most notably exchanges involving aromatic amino acids. We also observed differences in REs that appeared to be related to amino acid frequencies that were evident in model comparisons; those comparisons included models estimated using diverse proteins versus ribosomal proteins and models estimated using proteins from high-GC versus low-GC bacteria. Taken as a whole, these analyses demonstrate that REs estimated using the GTR_20_ model yield a meaningful picture of evolutionary model space for proteins.

## Figures and Tables

**Figure 1 biology-12-00282-f001:**
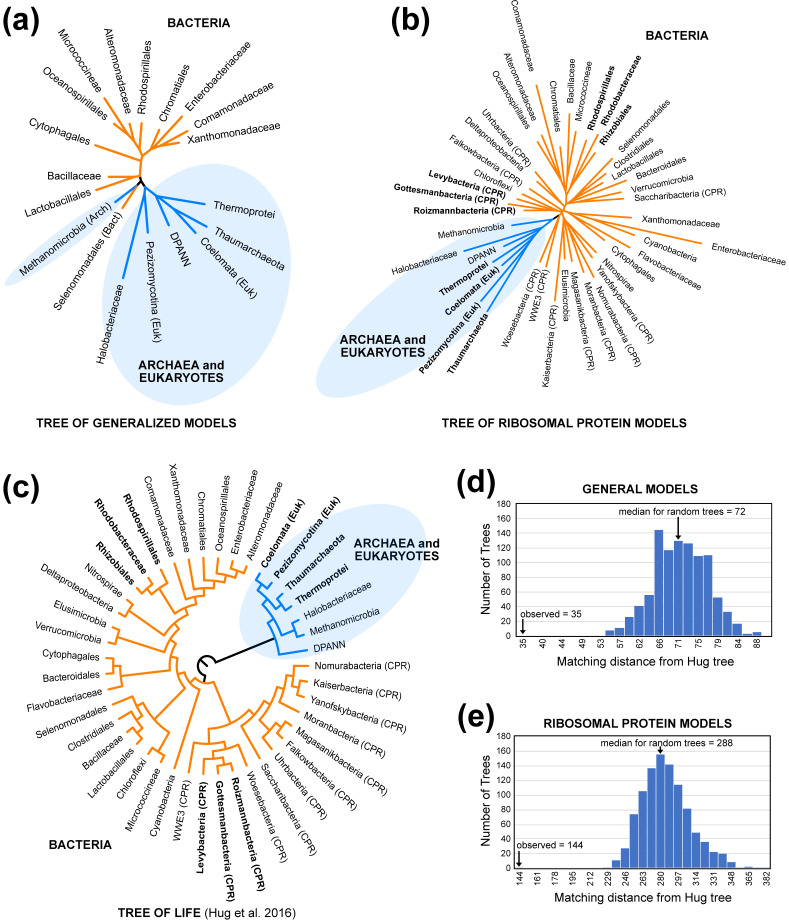
“Trees of models” and the tree of life exhibit topological similarities. The trees of models were generated by clustering Euclidean distances among estimated REs using neighbor-joining. Archaea and eukaryotes are shaded blue in all trees. (**a**) The tree of models for the general models. (**b**) The tree of models for ribosomal protein models. (**c**) Cladogram for the taxa used for this study based on the Hug tree, which is an ML analysis of concatenated ribosomal proteins [35]. “CPR” indicates members of the candidate phyla radiation [35,53,54]. (**d**) Matching distance between the Hug tree and the tree of general models compared with the distribution of distances between the Hug tree and random trees. (**e**) Matching distance between the Hug tree and the tree of ribosomal protein models compared with the distribution of distances between the Hug tree and random trees. The labels for each bin are the minimum matching distance for that bin.

**Figure 2 biology-12-00282-f002:**
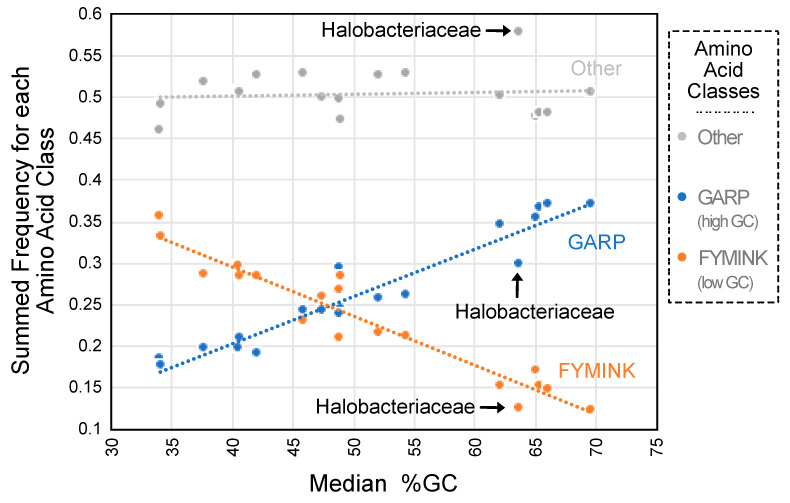
ML estimates of equilibrium amino acid frequencies are related to genomic GC content. The summed equilibrium frequencies for each amino acid class corresponded to values for the general models. We considered three amino acid classes: (1) GARP amino acids (glycine, alanine, arginine, and proline; encoded by GC-rich codons), (2) FYMINK amino acids (phenylalanine, tyrosine, methionine, isoleucine, asparagine, and lysine; encoded by GC-poor codons), and other amino acids. Summed equilibrium frequencies for GARP amino acids had a strong positive relationship with GC content (*f*_GARP_ = 0.0057[%GC] − 0.0245; r^2^ = 0.9222; indicated using the dashed blue line), whereas summed equilibrium frequencies for FYMINK amino acids had a strong negative relationship with GC content (*f*_FYMINK_ = −0.0059[%GC] + 0.5321; r^2^ = 0.9183; indicated using the dashed orange line). Sums of equilibrium frequencies for the remaining amino acids were unrelated to genomic GC content (indicated using the dashed gray line). Halobacteriaceae, which had the largest deviation from the line for other amino acids, is indicated on the graph (the deviation in the “other” category is positive, and this is likely the reason for the negative deviations in both the GARP and the FYMINK categories). Similar data for the individual amino acids involved in REs that exhibited large differences between high-GC and low-GC bacteria are presented in Appendix A.

**Figure 3 biology-12-00282-f003:**
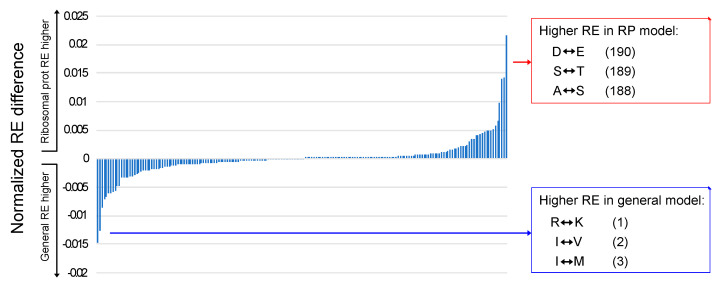
Comparison of REs for ribosomal protein models and general models. The normalized averages of the REs for general models were subtracted from REs for ribosomal protein models and sorted in ascending order. Large negative values correspond to higher REs in general models, and large positive values correspond to higher REs in the ribosomal protein models. The parenthetical numbers after each exchange correspond to the rank order of the differences when the RE differences are sorted from smallest to largest (i.e., 1 is the RE that was most elevated in the average general model relative to the average ribosomal protein model, 2 is the second most elevated, and so forth until 190, which are the REs that were most elevated in the average ribosomal protein model relative to the average general model). All RE estimates are available in Appendix A.

**Figure 4 biology-12-00282-f004:**
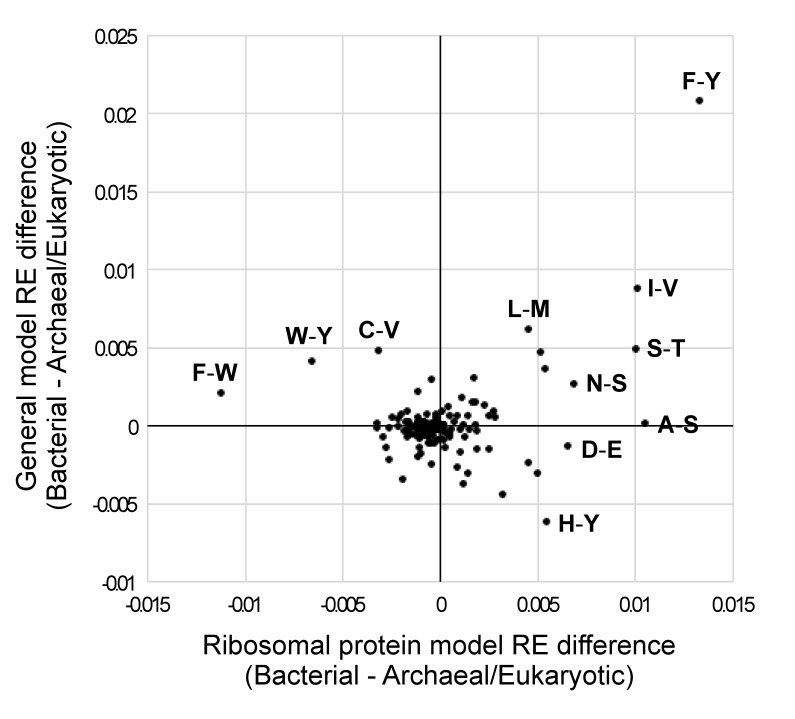
Comparison of RE difference values for bacteria versus archaea/eukaryotes for the ribosomal protein and general models. RE differences were generated by subtracting the average values for archaea + eukaryotes from those for bacteria. Archaea and eukaryotes were combined based on the near perfect bipartition (for generalized models) or perfect bipartition (for ribosomal protein models) that separated those taxa in our trees of models (Figure 1a,b). Positive values on each axis correspond to REs that were higher in the models trained on bacteria than in those trained using MSAs from archaea and eukaryotes; negative values were higher in the models trained on archaea + eukaryotes than in those based on bacterial MSAs. The REs for all models (along with these RE differences in this graph) are available in Appendix A.

**Figure 5 biology-12-00282-f005:**
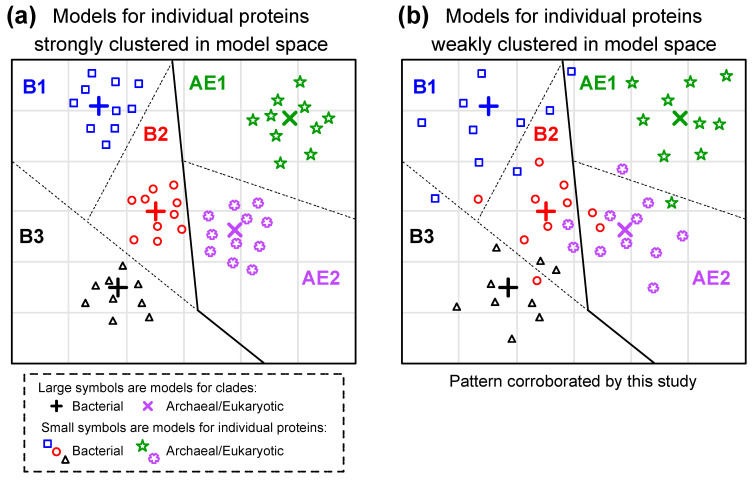
Potential patterns for variation among proteins and among clades in their underlying models of protein sequence evolution. Models for individual proteins are presented as either (**a**) strongly clustered or (**b**) weakly clustered. This conceptual illustration presents a two-dimensional model space. This was done to facilitate visualization; the actual model space defined by ***R*** matrices fore time-reversible 20 × 20 rate matrices has 190 dimensions. We only present five clades and ten proteins per clade to further simplify visualization. We divided model space into regions where the best-fitting model is the clade-specific model, with lines representing the boundaries of those regions (dotted lines divide regions within bacteria or archaea + eukaryotes; solid lines divide bacteria from archaea + eukaryotes). The results of this study support weak clustering in model space. In fact, the poor performance of models as classifiers (Table 1) suggests that models for individual proteins are very dispersed; we have limited the dispersal in this illustration to facilitate visualization.

**Table 1 biology-12-00282-t001:** Recall for the novel models when they were used as classifiers ^1^.

Domain	Clade	Median GC %	% Precise Match	% Archaeal Match	% Eukaryotic Match	% Bacterial Match
Archaea	DPANN	34.05	5.07	41.11	9.76	49.12
	Thaumarchaeota	34.20	25.12	37.68	5.68	56.64
	Methanomicrobia	47.40	18.50	55.49	7.52	36.99
	Thermoprotei ^2^	49.00	65.94	**76.56**	7.19	16.25
	Halobacteriaceae ^2^	63.70	65.39	**75.20**	4.86	19.94
Eukaryotes	Coelomata	40.72	10.74	50.93	15.96	33.11
	Pezizomycotina	48.90	14.80	33.42	39.63	26.95
Bacteria	Lactobacillales	37.70	18.90	24.56	12.01	63.43
	Cytophagales	40.60	13.24	10.29	7.35	**82.35**
	Bacillaceae	42.15	11.84	36.93	11.15	51.92
	Alteromonadaceae	45.90	7.33	13.56	6.31	**80.13**
	Selenomonadales	48.90	14.68	28.67	9.35	61.82
	Enterobacteriaceae	52.18	18.28	18.52	6.98	**74.50**
	Oceanospirillales	54.40	1.52	17.81	7.47	**74.72**
	Chromatiales	62.15	21.06	21.62	6.50	**71.88**
	Rhodospirillales	65.03	29.77	14.91	7.37	**77.72**
	Comamonadaceae	65.38	18.29	9.84	6.71	**83.46**
	Xanthomonadacae	66.03	9.62	11.70	10.78	**77.51**
	Micrococcineae	69.65	36.26	15.36	5.32	**79.32**

^1^ The % matches are the sums for matches to the relevant generalized models and the relevant ribosomal protein models. Values >70% are presented in bold. Full data are available in Appendix A. ^2^ The % precise matches for these taxa exceed 50%.

## Data Availability

Data used to train models and model parameters are available from Zenodo [47]. Model parameter files in the PAML format, which can be used by several phylogenetics programs (e.g., IQ-TREE), are also available from GitHub (https://github.com/ebraun68/protmodels, accessed on 20 December 2022).

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
