# Peer review of "The Structure of Evolutionary Model Space for Proteins across the Tree of Life"

_biology, 2023, doi:10.3390/biology12020282_

Round 1

Reviewer 1 Report

The most interesting idea in the paper, in my view, is to test if there is a topological congruence between the  tree of models and the tree of life. The presentation would be clearer if Figure 1 that presents the two would include some mapping between them, allowing the reader to easily see the similar parts. Also with respect to Figure 1, it is unclear why when relying on the ribosomal proteins there are so many more organisms (I would assume it would be the same number) (comparing 1a and 1b). Also, it is unclear why the random distribution for the two differs (comparing 1d and 1e).

The studies of Loren Williams relating RNA proteins and evolution are not cited, and are very relevant. 

Author Response

Responses:

  1. The reviewer suggests that figure 1 should be modified to facilitate comparisons among trees. Although we agree that it is important to present trees in ways that facilitate comparisons, we highlighted the bipartition separating bacteria from archaea+eukaryotes using both color and shading. We also bolded clade names for the subgroups in the ribosomal protein model tree that match the topology of the species tree (Figure 1c). We feel it would be difficult to further modify the presentation of the trees of models while still maintaining the use of unrooted trees, which we believe to be important because we want to emphasize that the clustering diagrams (“trees of models”) are not estimates of phylogeny. Instead, the goal of our study was to determine whether or not the trees of models were similar to the tree of life. Therefore, we would like to keep the figure as is.
  2. The reviewer asked why we used more ribosomal protein models than generalized models. As we stated in the methods we initially selected 19 clades, selecting clades with numbers of taxa that were convenient for the methods we used. Ffr the ribosomal proteins we felt that it would be possible to expand this number of clades without greatly increasing the computational burden, so we chose to do so. But this was because there were fewer aligned sites in the ribosomal protein data. Ultimately, any study of this type will face the question of taxon sampling - how many taxa should be included? At present, there are >500,000 genome sequences available for analysis, so some sort of choices will have to be made. We feel we chose a reasonable set of clades that is spread out across the tree of life and prefer to keep it as is.
  3. The reviewer asked why the distributions of matching distances between random trees and the tree of life differed for the trees of general models and ribosomal protein models. This is a simple reflection of the different numbers of OTUs in the trees. We have modified the text to indicate this.
  4. The reviewer suggests that we comment on the work of Loren Williams in the manuscript. We have added a citation to a review for the Williams lab (Bowman, J.C.; Petrov, A.S.; Frenkel-Pinter, M.; Penev, P.I.; Williams, L.D. Root of the tree: the significance, evolution, and origins of the ribosome. Chem. Rev. 2020, 120, 4848–4878) in section 3.8. Reading the Bowman et al. review led us to additional literature on the evolution of the ribosome, and that reading prompted us to propose a potential relationship between patterns of evolution for the ribosome exit tunnel and the bipartition separating Bacteria from Archaea+eukaryotes in our tree of models. We spent part of a paragraph briefly articulating this idea. We thank the reviewer for this idea that enriched the manuscript.

Reviewer 2 Report

The work entitled “The structure of evolutionary model space for proteins across the tree of life” by Scolaro and Braun projects a very interesting story on Protein evolution or more precisely proteome evolution across the life forms based on amino acid substitution on evolutionary time. Relative rates of amino acid substitution can be used as a proxy for the evolutionary preference of amino acids with respect to their physico-chemical properties and in the context of specific proteins and their biological relevance. Authors in their present story presented the evolution across the tree of life and have showed that bacterial models differed form archaeal and eukaryotic models; more precisely models for Halobacteriaceae and Thermoprotei were very distinctive. The work in general very nicely describes the “evolutionary model space”. But for the general readership the work needs some additional elements which the authors may consider for the subsequent revised version of the draft.

1.     Authors have shown that Genomic GC content had a modest impact on relative exchangeabilities despite having a large impact on amino acid frequencies. It would be interesting if they correlate this information with protein conformation and super-secondary organizations.

2.     Authors have mentioned “The bacterial and archaeal clades with relatively high recall (Halobacteriaceae and Micrococcineae) also have the highest genomic GC content (Table 1).” Authors have discussed this point in the context of Genomic base composition and how it affects amino acid frequencies and relative exchangeability. It would be interesting to investigate how these genomic base compositions vary across the protein families which are in particular critical for the evolvability. Authors can be selective in this aspect, but this would allow us to be more specific to monitor protein evolution as opposed to general inference about the differential GC contents.

3.     As stated in point 1 genomic base composition upon correlation with protein conformation would be interesting to investigate. It would be interesting to see if there is some trend towards secondary structural propensities and how that varies across the tree of life and how well it correlates with the general trend which the authors have observed.

4.     The study in general emphasizes on whole proteome; authors can include few specific examples of specific proteins which are critical to evolvability and see how the trend in RE is reflected. Furthermore, authors have mentioned that evolutionary trajectories and sequence relative exchangeability exhibits historical (evolutionary) and ecological signature. This would be interesting to note in the context of specific proteins. Authors can test out few selective proteins and see how the trend emerge.

5.     It is encouraging to note that the Tree of models showed similarities to tree of life. Has the authors tried to do some covariational studies. Authors can include protein-protein interaction covariational studies along with single protein covariation studies to see if the RE associated sequence variations or positions associated are under evolutionary constraint and if those points are coevolutionary contacts.

Overall, the work is very interesting, and the analyses have been carried out thoroughly. But to be of interest to broad readership of protein structure and evolution the article needs some refinements. Moreover, the message of the paper and of the individual sections are lost many times. This could be because authors have focused on the method. This needs to be addressed by carefully rephrasing sections and the discussion so that it can be interest to the broader readership.

Author Response

***

GENERAL RESPONSE:

This reviewer makes a number of very interesting points. Indeed, upon reading this review I found myself wishing I could have a long discussion with the reviewer because the points they articulate provide excellent ideas for follow up studies. Fundamentally, many of this reviewer’s suggestions are that we drill down into specific genes or groups of genes. 

Ultimately, we feel that the best way to address most of the comments is to provide additional explanation for the issues highlighted by the reviewer. In many cases, the reviewer’s advice is to present more details about specific proteins or groups of proteins. We do present some information about ribosomal proteins. However, I believe that addressing many of the reviewer’s comments in a detailed manner would expand the manuscript in a way that fundamentally changes the focus of the manuscript. Many of the ideas in this review would be very interesting for follow up studies that examine patterns of sequence evolution in specific protein families. 

*** 

  1. Authors have shown that Genomic GC content had a modest impact on relative exchangeabilities despite having a large impact on amino acid frequencies. It would be interesting if they correlate this information with protein conformation and super-secondary organizations.

***

RESPONSE:

This is an interesting idea. A number of researchers in the field (including the senior author) have worked on methods to integrate evolutionary models with protein structure. However, given the modest impact of genomic GC-content on relative exchangeabilities (REs), which were the focus of our manuscript, any attempt to explore additional consequences of genomic GC-content would take us off topic.

Moreover, we want to stress that this manuscript uses a method that does not incorporate structural information: simply calculating 20x20 rate matrices. Although the use of 20x20 rate matrices has a long history it remains commonly used (a recent paper in this area by another is: Minh, B.Q.; Dang, C.C.; Vinh, L.S.; Lanfear, R. Qmaker: fast and accurate method to estimate empirical models of protein evolution. Syst. Biol. 2021, 70, 1046–1060, doi:10.1093/sysbio/syab010). Thus, we do not feel the method is out of date - it is simply a straightforward method. Moreover, we believe the positive results we obtained using the simple method show that it is useful to answer the questions we posed. 

***

  1. Authors have mentioned “The bacterial and archaeal clades with relatively high recall (Halobacteriaceae and Micrococcineae) also have the highest genomic GC content (Table 1).” Authors have discussed this point in the context of Genomic base composition and how it affects amino acid frequencies and relative exchangeability. It would be interesting to investigate how these genomic base compositions vary across the protein families which are in particular critical for the evolvability. Authors can be selective in this aspect, but this would allow us to be more specific to monitor protein evolution as opposed to general inference about the differential GC contents.

  1. As stated in point 1 genomic base composition upon correlation with protein conformation would be interesting to investigate. It would be interesting to see if there is some trend towards secondary structural propensities and how that varies across the tree of life and how well it correlates with the general trend which the authors have observed.

***

RESPONSE:

Points 2 and 3 are related to the potential for correlations between genomic GC content and various aspects of the proteome. This is an interesting area, but, as we stated above, we ultimately found that genomic GC-content has a modest impact on REs. Since REs were the core focus of our manuscript we don’t want to expand our discussion of the topic.

***

  1. The study in general emphasizes on whole proteome; authors can include few specific examples of specific proteins which are critical to evolvability and see how the trend in RE is reflected. Furthermore, authors have mentioned that evolutionary trajectories and sequence relative exchangeability exhibits historical (evolutionary) and ecological signature. This would be interesting to note in the context of specific proteins. Authors can test out few selective proteins and see how the trend emerge.

  1. It is encouraging to note that the Tree of models showed similarities to tree of life. Has the authors tried to do some covariational studies. Authors can include protein-protein interaction covariational studies along with single protein covariation studies to see if the RE associated sequence variations or positions associated are under evolutionary constraint and if those points are coevolutionary contacts.

***

RESPONSE:

Points 4 and 5 are requests to shift the focus from our large-scale analysis of the proteome to describe patterns of sequence evolution for specific protein families (and sites involved with protein-protein interactions in the case of point 5). We certainly agree that this is an interesting area to pursue, but, we also feel that it would take us away from our focus on proteome-wide REs. We feel that moving beyond the comparison of ribosomal proteins and the proteome-wide mean RE values would dilute the focus of the manuscript.

***

Overall, the work is very interesting, and the analyses have been carried out thoroughly. But to be of interest to broad readership of protein structure and evolution the article needs some refinements. Moreover, the message of the paper and of the individual sections are lost many times. This could be because authors have focused on the method. This needs to be addressed by carefully rephrasing sections and the discussion so that it can be interest to the broader readership.

***

RESPONSE:

We have tried to make sure that section titles describe the contents of each section. We have also added a figure (Figure 5) that provides a very simplified visualization of our recall data. We hope this is helpful to the reviewer and to readers.

***

Reviewer 3 Report

The study by Scolaro and Braun “The structure of evolutionary model space for proteins across the tree of life” explores the fitting of a tree based on substitution models of protein evolution from different taxa with the tree of life, including certain signatures of molecular evolution like the GC content, distribution of amino acid frequencies and different types of amino acid substitutions. The results, “a small number of relative exchangeabilities are responsible for much of the structure of the model space for protein sequence evolution”, are highly surprising. In general, I found that the study is interesting to confirm current knowledge. However, some methods require clarification and some aspects written in the manuscript should be clarified and increased. Altogether I recommend revisions.

Major comments

The relevance of this study in not clear in the manuscript. The last line of the abstract suggests that this study can be used for phylogenomic inferences. However, how this could be possible is not mentioned anymore along the manuscript.

Also, related with the cited application of this study, why substitution models are useful in phylogenetics/phylogenomics considering the following studies?

https://academic.oup.com/mbe/article/37/7/2110/5810088

https://www.nature.com/articles/s41467-019-08822-w

At least this aspect should be discussed.

In the third paragraph of the Introduction is mentioned that “Individual proteins will not provide enough information to allow accurate estimation of GTR20 model parameters”, which is far from true. A number of studies inferred substitution models for a particular protein considering only evolutionary information from such a protein. This was done through both empirical and parametric substitution models. For example in https://www.mdpi.com/2073-4425/13/1/61 authors inferred empirical substitution models for specific HIV-1 proteins that fitted better with the data than models based on many proteins (including models based on only HIV-1 proteins), showing that protein-specific substitution models are more realistic than substitution models based on many different proteins (that probably were subjected to different types of selection). Also parametric substitution models, based on evolutionary information from the protein structure, showed a much better fitting than the substitution models based on sequences from many proteins, see for example,

https://academic.oup.com/mbe/article/31/3/736/1008553?login=false

https://academic.oup.com/mbe/article/32/8/2195/2925560?login=false

If one has a sufficient number of sequences of the query protein (i.e., the first study cited in this comment, thousands of sequences are available in databases for every HIV protein) then it is possible to accurately infer a reversible exchangeability matrix. Besides, one can use information from the protein structure to infer substitution models without requiring a high number of sequences. Next, since selective pressures can differ among genes of the same genome leading to different substitution models that best fit with different genes (i.e., https://academic.oup.com/gbe/article/doi/10.1093/gbe/evr080/591755?login=false), it is not surprising that substitution models for different proteins also differ. Moreover, selective pressures often differ among sites or regions of a protein (i.e., the selection pressure in sites of the active center of an enzyme is completely different to that in sites of the enzyme surface) and thus the best fitting substitution model can differ among protein sites. For example, https://journals.plos.org/plosbiology/article/comments?id=10.1371/journal.pbio.1002452

I understand that this study focuses on the tree of life for which is convenient to infer and study generalist substitution models, but it should be clear that those models are very unrealistic because combine information from very different proteins that could evolve under different evolutionary constraints. I think these aspects should be mentioned and discussed in the manuscript.

In Materials and Methods, more information is needed to justify the choice of the analyzed data. For example, why only those 19 different clades? Are they sufficient? Or in “We collected files of annotated protein sequences for 11 to 37 taxa in each clade”, is this number of taxa sufficient? Note that empirical substitution models of protein evolution are usually based on thousands of sequences.

Another important issue in this study is obtaining accurate multiple sequence alignments (MSAs). Since one is aligning sequences with so much diversity, what is the error of the MSA of this study? Which penalties for opening gaps and gaps length were used?, note that aligning data from different species can be problematic, many regions in a species may not be in another species and homology could be lost in certain regions. How indels were treated? If indels are removed, what is the final length of the study data? Is the final data enough representative?

Another important issue in this study is recombination. Recombination and other processes of exchange of genetic material can bias phylogenetic tree reconstructions, https://academic.oup.com/genetics/article/156/2/879/6051420?login=false

The tree of life is really a network of life because many processes involved transfer and exchange of genetic information among species, https://www.ncbi.nlm.nih.gov/pmc/articles/PMC2874017/

Thus, the tree of life is an approximation that should in mind if one aims to directly compare it with other aspects like the tree of models. This should be at least discussed in the manuscript.

The study presents some trees of models. I am wondering if those trees can include any statistical support like bootstrap values to see how much trustable are the internal nodes (divergence among models). Is the error lower than the estimate?

The conclusions of the study are expected in my view. It is well known that the different kingdoms of life have numerous different molecular properties. For example, different genetic codes (i.e., comparing bacteria and archaea, mitochondrial and nuclear genes, etc). And also, on the other side, using general substitution models of protein evolution (where each model is based on the evolution of many different proteins) that combine different evolutionary signatures can result in a poor signal of the evolutionary histories. So I think the manuscript should more clearly mention the goal of this study, for example that it can be useful for phylogenomic reconstructions (see also the first main comment).

Minor comments

At the beginning of the introduction I missed at least a reference, i.e. a review, about the meaning and types of substitution models of evolution that could be useful for non-experts. I remember the following, https://www.ncbi.nlm.nih.gov/pmc/articles/PMC4620419/

The following sentence would require support by a some references “Evidence that REs exhibit a historical signal (i.e., patterns in the data that are congruent with at least some parts of the tree of life), an ecological signal (i.e., strong differences between RE estimates for taxa living in distinct environments, like thermophiles versus mesophiles), or both signals would indicate the violations of local time-reversibility are not especially problematic.”

I think the manuscript should be more cautious in many categorical sentences. For example, at end of the discussion “Moreover, a fundamental problem for all efforts to improve the realism of models of sequence evolution is that they may require the addition of free parameters and it is possible that some of those parameters will be non-identifiable (or only weakly identifiable; see Ponciano et al. [66]  for discussion).” Also it is possible to consider identifiable parameters that can be used to improve the fitting with the data (i.e., see refs in the second major comment). So that sentence is confusing and sounds biased because the opposite is also possible but not mentioned.

Author Response

***

GENERAL RESPONSE:

Like the second reviewer, this reviewer makes a number of very interesting points. In this case, the reviewer has focused on the issue of tree estimation. Obviously, tree estimation is an important and very broad area of research. Indeed, there are entire journals focused on the research area (for example, a relatively large proportion of papers published in some high-impact journals, like Systematic Biology, deal with issues related to tree estimation). However, tree estimation is not the core focus of this paper; the primary focus is on understanding the process of protein evolution using the lens of relative exchangeabilites (REs) of amino acids.

With that said, we understand that many readers may be interested in tree estimation. Moreover, we do believe that our models have the potential to have a positive impact on phylogenetic estimation (although we have added some much needed nuance to that assertion based on this reviewer’s comments). However, this is a side issue, in our opinion. We have added two sections to this manuscript and added a figure (Figure 5) to address this reviewer’s comments. We have also rewritten the introduction, taking this reviewer’s advice to clarify the point of the manuscript. We believe that these changes improve the clarity of the manuscript and address the reviewer’s criticisms. 

We would also like to stress that the final paragraph of the new introduction lists three major questions very explicitly:

“First, what type(s) of signal(s) are present in clade-specific model REs? In principle, there could be a phylogenetic signal, an ecological signal, or some mixture of both types of signals.” 

“Second, do the best models for individual proteins cluster near a taxon-specific mean?” 

“Third, which clade-specific model pa-rameters exhibited differences?”

We also describe the ways that we answer each of these questions after we state each question. We hope that the rewritten introduction clarifies the manuscript. See specific responses below for additional details. None of these questions address phylogeny directly (with the exception the statement that signal in the RE data *could* be phylogenetic). We hope that this rewritten introduction focuses the reviewers and readers on the major goals of the manuscript.

We respond to specific comments in more detail below, separated from the reviewer's comments using asterisks and "RESPONSE." We hope that our additions to the manuscript improve the clarity and ultimately make the manuscript more valuable to readers.

*** 

Major comments

The relevance of this study in not clear in the manuscript. The last line of the abstract suggests that this study can be used for phylogenomic inferences. However, how this could be possible is not mentioned anymore along the manuscript.

Also, related with the cited application of this study, why substitution models are useful in phylogenetics/phylogenomics considering the following studies?

https://academic.oup.com/mbe/article/37/7/2110/5810088

https://www.nature.com/articles/s41467-019-08822-w

At least this aspect should be discussed.

***

RESPONSE:

The primary goal of this manuscript was to understand patterns of sequence evolution, not to obtain a better estimate of the tree of life. We did make the following comment in the abstract:

“If we look beyond the information that these clade-specific models reveal about protein evolution the models themselves are likely to be useful tools for phylogenomic inference across the tree of life.”

We admit that this may have been a bit confusing since our goal was to understand the models, not to improve estimates of phylogenetic trees. We do want to stress that most of the original abstract described insights from the models (e.g., the fact that the structure of model space was probably shaped by phylogeny, that specific RE values tended to vary among models, and so forth). Ultimately, this was simply a bit of clumsy writing that appears to have caused the reviewer to focus on the wrong aspects of the manuscript. To improve the abstract we changed the sentence above to read:

“The clade-specific models we generated may be useful tools for protein phylogenetics and the structure of evolutionary model space that they revealed has implications for phylogenomic in-ference across the tree of life.”

We feel that this shifts the focus to the implications of the models and changes the phrasing “...are likely to be useful tools…” to the more modest “...may be useful…”. 

Returning to the two papers the reviewer highlighted above, we feel that the most relevant is the first (Spielman, S.J. Relative model fit does not predict topological accuracy in single-gene protein phylogenetics. Mol. Biol. Evol. 2020, 37, 2110–2123, doi:10.1093/molbev/msaa075). The other paper is interesting, but it focuses on nucleotide models and therefore it is not directly relevant to this manuscript, despite having a conclusion essentially identical to Spielman (2020). I was aware of Spielman (2020) but did not feel it was directly related to the present manuscript. Given the reviewer’s comments I added a section to the manuscript (Section 3.9, entitled “Can our clade-specific models improve estimates of the tree of life?”) and included a citation of Spielman (2020) in that section.

I believe it is difficult to use the conclusions of Spielman (2020) in practice, so I am not completely sure how they can be addressed. This is because Spielman (2020) concludes that model fit does not improve tree topology for single protein analyses. However, using the best-fit model did not degrade the performance of phylogenetic estimation. So, the conclusion is really that model doesn’t matter. My concern with that conclusion is that one still needs to use *some* model for ML tree estimation. It is one thing to say that one doesn’t have to worry too much about the specific model one uses (and I believe that would be a perfectly reasonable “takeaway” from Spielman 2020) but it seems unreasonable to assert that one shouldn’t pick the best-fitting model if one tests multiple models. 

One potential “action plan” based on Spielman (2020) would be to say that one should use a single model for all analyses (perhaps the commonly used LG model or the more recent Q.pfam model) or that one should use the GTR20 model and optimize all parameters. But it is possible that there are parts of parameter space where LG and Q.pfam perform poorly. The question of whether or not model fit can detect these problematic parts of parameter space is an open one. One could optimize all GTR20 parameters, but one can also test a large number of simple 20x20 matrix models quite rapidly. If a number of those models are available, why not just screen those models? It is probably faster to do that than to optimize all parameters of the GTR20 model, since testing the likelihood of a simple 20x20 matrix model would only require optimizing rate heterogeneity and branch length parameters but optimizing GTR20 parameters requires optimizing all of those parameters *plus* all of the rate matrix parameters.

We recognize, of course, that the results of Spielman (2020) do raise questions about the value of adding models to the existing set of models. In fact, if the only reason that we were adding models was to provide a resource for tree estimation we would agree that Spielman (2020) might undermine the value of adding those models. We would also temper that statement by saying that we would like to see additional studies that came to the same conclusions as Spielman (2020). This is not an assertion that Spielman (2020) is wrong; upon rereading Spielman (2020) it appeared to be a good study. However, it was still dependent on the details of a specific simulation scheme. Spielman (2020) did not simulate data using a simple 20x20 matrix; that was an appropriate strategy since simulations using a simple 20x20 matrix would very likely yield biased conclusions. However, it is far from clear that Spielman (2020)’s conclusions would hold for a larger set of “reasonable” models of protein evolution. Overall, it seems to me that embracing Spielman (2020)’s conclusions and taking them to their logical conclusions is a statement that many papers from multiple labs, including some very recent publications (e.g., Dang, C. C., & Vinh, L. S. 2023. Estimating amino acid substitution models for metazoan evolutionary studies. J.Evol. Biol, online ahead of print, doi: 10.1111/jeb.14147) are misguided. As scientists, we should always be open minded regarding the possibility that any specific approach we are using *is* misguided, but we also feel that more than one or two papers are needed to prove that contention!

Moreover, we did not estimate models as a resource for tree building; we estimated for the express purpose of comparing the models, looking at their ability to identify the clade of origin for alignments (i.e., to examine the behavior of models as classifiers), and understanding which of the RE values vary across taxa. These are the three questions we ask at the end of our introduction (see our general response above). Now that the models exist we feel it is possible that they will be useful for phylogenetics. We acknowledge that, if Spielman (2020) is correct and models really don’t matter for tree estimation, then the new models may not be that useful for tree estimation. However, we feel that the results we present indicate that the models we estimated were useful for answering the questions that we asked in the introduction.

*** 

In the third paragraph of the Introduction is mentioned that “Individual proteins will not provide enough information to allow accurate estimation of GTR20 model parameters”, which is far from true. A number of studies inferred substitution models for a particular protein considering only evolutionary information from such a protein. This was done through both empirical and parametric substitution models. For example in https://www.mdpi.com/2073-4425/13/1/61 authors inferred empirical substitution models for specific HIV-1 proteins that fitted better with the data than models based on many proteins (including models based on only HIV-1 proteins), showing that protein-specific substitution models are more realistic than substitution models based on many different proteins (that probably were subjected to different types of selection). Also parametric substitution models, based on evolutionary information from the protein structure, showed a much better fitting than the substitution models based on sequences from many proteins, see for example,

https://academic.oup.com/mbe/article/31/3/736/1008553?login=false

https://academic.oup.com/mbe/article/32/8/2195/2925560?login=false

If one has a sufficient number of sequences of the query protein (i.e., the first study cited in this comment, thousands of sequences are available in databases for every HIV protein) then it is possible to accurately infer a reversible exchangeability matrix. Besides, one can use information from the protein structure to infer substitution models without requiring a high number of sequences. Next, since selective pressures can differ among genes of the same genome leading to different substitution models that best fit with different genes (i.e., https://academic.oup.com/gbe/article/doi/10.1093/gbe/evr080/591755?login=false), it is not surprising that substitution models for different proteins also differ. Moreover, selective pressures often differ among sites or regions of a protein (i.e., the selection pressure in sites of the active center of an enzyme is completely different to that in sites of the enzyme surface) and thus the best fitting substitution model can differ among protein sites. For example, https://journals.plos.org/plosbiology/article/comments?id=10.1371/journal.pbio.1002452

I understand that this study focuses on the tree of life for which is convenient to infer and study generalist substitution models, but it should be clear that those models are very unrealistic because combine information from very different proteins that could evolve under different evolutionary constraints. I think these aspects should be mentioned and discussed in the manuscript.

***

RESPONSE:

We would like to start with the following statement by the reviewer:

“I understand that this study focuses on the tree of life for which is convenient to infer and study generalist substitution models, but it should be clear that those models are very unrealistic because combine information from very different proteins that could evolve under different evolutionary constraints. I think these aspects should be mentioned and discussed in the manuscript.”

We completely agree, but we also feel that, as the late George Box said, “all models are wrong but some models are useful.” Single rate matrix models of the type we used, with REs averaged over many proteins, are almost certainly less realistic than, for example, than a model that models among sites variation in amino acid profiles using a Dirichlet process prior (i.e., the Bayesian CAT model; Lartillot, N.; Philippe, H. A Bayesian mixture model for across-site heterogeneities in the amino-acid replacement process. Mol. Biol. Evol. 2004, 21, 1095–1109, doi:10.1093/molbev/msh112). But the CAT model is itself far from perfect (this is not to criticize CAT; it is clearly useful in some contexts).

Thus, we feel the question is not whether or not the set of single rate matrix models that we used are realistic; it is whether they are useful. In other words, we believe that the contention that a specific model is useful should be viewed as a hypothesis. Then the question becomes: can we *corroborate* the *hypothesis* that single rate matrix models are useful to answer the questions posed by any specific study? 

We actually articulated criteria to judge whether or not our models were useful. Specifically, we wanted to determine two things about the models: 1) could be used as classifiers?; and 2) would clustering distances among model parameters yield a tree with interpretable groups? The answer to the first question was: 1) the models could be used as precise classifiers in two cases (Halobacteriaceae and Thermoprotei); and 2) they could be used as domain-level classifiers in 11 out of 19 cases. The answer to the second is that clustering distances among models yields a tree that is more similar to an estimate of the tree of life than expected by chance (Figure 1d,e). Although one must be cautious when focusing on individual tree bipartitions unless they are specified a priori, we also note that trees of models supported a separation between archaea+eukaryotes and bacteria (albeit imperfectly in the case of the generalized models). This is arguably the most prominent bipartition in the tree of life.

It is certainly possible that “better” models might improve those results; it is an almost universal truism that there is room to improve models used for inference in science.. However, the nature of those improved models is far from clear. The reviewer generally seems focused on variation among individual proteins, and our results actually show that there is substantial variation. But the precise mathematical framework that would allow us to address the questions we asked while incorporating variation among proteins in a principled way is unclear. We have added a section (Section 3.7, Does our approach provide an accurate picture of protein evolutionary model space?) that discusses the issues the reviewer raises. Hopefully, this will be useful to readers.

Moving on, we have removed the explicit statement that individual proteins do not have enough information to parameterize the GTR model. 

We agree with the reviewer’s comment that individual proteins differ in their patterns of evolution. We approached the issue of variation among proteins in a way that differs from the way the reviewer appears to advocate. Specifically, we defined a set of models and then asked which model was the best-fitting model for individual proteins from a specific clade. Our group has used this strategy to analyze proteins from several eukaryotic groups (Pandey, A.; Braun, E.L. Protein evolution is structure dependent and non-homogeneous across the tree of life. In Proceedings of the 11th ACM International Conference on Bioinformatics, Computational Biology and Health Informatics, Virtual Event, 21–24 September 2020; ACM: New York, NY, USA, 2020; pp. 1–11. doi:10.1145/3388440.3412473). B.Q. Minh, R. Lanfear, and colleagues used a virtually identical strategy (using GTR20: Minh, B.Q.; Dang, C.C.; Vinh, L.S.; Lanfear, R. Qmaker: fast and accurate method to estimate empirical models of protein evolution. Syst. Biol. 2021, 70, 1046–1060, doi:10.1093/sysbio/syab010; using the unrestricted 20-state model: Dang, C.C.; Minh, B.Q.; McShea, H.; Masel, J.; James, J.E.; Vinh, L.S.; Lanfear, R. nQMaker: estimating time non-reversible amino acid substitution models. Syst. Biol. 2022, 71, 1110–1123, doi:10.1093/sysbio/syac007). All of those studies showed that models for most individual proteins were clustered near clade-specific means. 

The reviewer appears to be advocating a different strategy: simply optimizing all parameters for the GTR20 model. This may be a very reasonable strategy and comparing the approach we used to the direct parameter optimization approach would be very interesting as a follow up study. However, a benefit of the strategy that we used is that we can compare our results to the results of analyses using data from several different eukaryotic clades (Pandey and Braun 2020, Minh et al. 2021, and Dang et al. 2022). We found that individual proteins appeared to exhibit greater “scatter” relative to the clade-specific mean when prokaryotic proteins were used, in sharp contrast to the results of previous studies. We also found different degrees of model scatter across clades (e.g., Halobacteriaceae and Thermoprotei had less scatter than many other prokaryotic groups). It may end up being the case that direct estimates of GTR20 parameters perform as well as the approach we used, but we do not have the baseline comparisons. We feel that the background information from the published papers from the lab of the senior author on this and from Minh+Lanfear lab is valuable and that it justifies our decision.

Finally, we agree with the reviewer that information from protein structure can be very useful. However, we did not use structure in this study. It would be more appropriate for a follow up study.

*** 

In Materials and Methods, more information is needed to justify the choice of the analyzed data. For example, why only those 19 different clades? Are they sufficient? Or in “We collected files of annotated protein sequences for 11 to 37 taxa in each clade”, is this number of taxa sufficient? Note that empirical substitution models of protein evolution are usually based on thousands of sequences.

***

RESPONSE:

We used 1000 proteins to estimate models in all but one case:

“The training sets for all clades except Thermoprotei comprised 1000 MSAs (we obtained fewer clusters from Thermoprotei so that training set was limited to 640 MSAs).” (lines 213-215)

We did limit the taxon sample. However, we wanted to examine shifts in the models across the tree of life and we wanted to use Hug et al.(Hug, L.A. et al.. A new view of the tree of life. Nat. Microbiol. 2016, 1, 16048, doi:10.1038/nmicrobiol.2016.48) to guide our taxon selection. This allowed us to compare our general models with our ribosomal protein models (using Hug et al. as the source of the ribosomal protein sequences).

We understand the need for a large number of taxa when one is using a single protein (e.g., the case in Del Amparo, R.; Arenas, M. HIV protease and integrase empirical substitution models of evolution: Protein-specific models outperform generalist models. Genes 2021, 13, 61, doi:10.3390/genes13010061, which the reviewer cites below). However, using a large number of protein MSAs with fewer taxa is a perfectly reasonable alternative. After all, the number of amino acid changes in a large number of MSAs with few taxa is likely to be similar to a single MSA with many taxa. Moreover, the use of many MSAs is more consistent with the goal of this study. After all, we *wanted* estimates of REs averaged over many proteins.

***

Another important issue in this study is obtaining accurate multiple sequence alignments (MSAs). Since one is aligning sequences with so much diversity, what is the error of the MSA of this study? Which penalties for opening gaps and gaps length were used?, note that aligning data from different species can be problematic, many regions in a species may not be in another species and homology could be lost in certain regions. How indels were treated? If indels are removed, what is the final length of the study data? Is the final data enough representative?

***

RESPONSE:

We did not alter the alignments generated by usearch in any way, so gaps were retained. This is now stated in the methods. 

***

Another important issue in this study is recombination. Recombination and other processes of exchange of genetic material can bias phylogenetic tree reconstructions, https://academic.oup.com/genetics/article/156/2/879/6051420?login=false

The tree of life is really a network of life because many processes involved transfer and exchange of genetic information among species, https://www.ncbi.nlm.nih.gov/pmc/articles/PMC2874017/

Thus, the tree of life is an approximation that should in mind if one aims to directly compare it with other aspects like the tree of models. This should be at least discussed in the manuscript.

***

RESPONSE:

We do not feel that recombination (horizontal transfer) is likely to affect our results. The method we used to estimate REs optimizes model parameters over a set of trees, so any discordance among gene trees would be accommodated in the estimation method. This is now stated in the methods.

One of the sections we added in response to this reviewer (section 3.9. Can our clade-specific models improve estimates of the tree of life?) also makes the point that supertree methods that combine trees often yield estimates of the tree of life with some similarities to the Hug tree, which we used as our baseline estimate of the tree of life. We decided to add a version of the most comprehensive supertree (the tree from Zhu, Q., et al.. Phylogenomics of 10,575 genomes reveals evolutionary proximity between domains Bacteria and Archaea. Nat. Commun. 2019, 10, 5477, doi:10.1038/s41467-019-13443-4) pruned to match our taxon sample as a supplementary figure (Supplementary Figure S2). 

We thank the reviewer for pointing out this potential problem; it allowed us to add information arguing that conclusions based on our methods are likely to be robust to horizontal transfer.

***

The study presents some trees of models. I am wondering if those trees can include any statistical support like bootstrap values to see how much trustable are the internal nodes (divergence among models). Is the error lower than the estimate?

***

RESPONSE:

This is an interesting idea. Obtaining statistical support in a manner similar to the bootstrap  might be possible, but it would probably require changes to our methodology. We chose to examine support for our tree by asking whether the matching distance between the Hug tree (which we treated as the tree of life) and our tree of models was less than expected for random trees (this is shown in Figure 1d,e and the methods are presented in section 2.2). The matching distance analysis provides evidence that the tree of models is significantly closer to the Hug tree than expected by chance, corroborating the hypothesis that the RE data exhibits a phylogenetic signal. However, the method we used does not provide support values for specific branches.

However, we want to stress that the tree of models is not an estimate of phylogeny. It was a method to test whether there was a phylogenetic signal in the data. In principle, the signal in the data might have been exclusively ecological in nature. For example, if thermophiles had been grouped but archaea and bacteria had been intermixed we would have concluded that there was no phylogenetic signal (more accurately, we would have concluded that, if there was a phylogenetic signal in the mean RE values that signal was too weak to be detected using our methodology). Ultimately, the evidence from the matching distance analysis indicates that there is a phylogenetic signal in the model data, but that result was not guaranteed. However, it is important to note that broader sampling of clades *might* reveal a mixture of phylogenetic and ecological signals. With that said the evidence for a phylogenetic signal in these data indicate that any ecological signal would be superimposed on a phylogenetic signal.

***

The conclusions of the study are expected in my view. It is well known that the different kingdoms of life have numerous different molecular properties. For example, different genetic codes (i.e., comparing bacteria and archaea, mitochondrial and nuclear genes, etc). And also, on the other side, using general substitution models of protein evolution (where each model is based on the evolution of many different proteins) that combine different evolutionary signatures can result in a poor signal of the evolutionary histories. So I think the manuscript should more clearly mention the goal of this study, for example that it can be useful for phylogenomic reconstructions (see also the first main comment).

***

RESPONSE:

It is difficult to respond to this comment since it seems to be a statement that the results of the study are trivial and expected. Obviously, we would argue the opposite. To the extent that the results of this study conflict with those of prior studies using similar methods and eukaryotic data (Pandey and Braun 2020, Minh et al. 2021, and Dang et al. 2022, cited above) they are unexpected. In fact, the reviewer seems to be arguing that they expect the dominant pattern of variation in protein models to be variation among individual proteins; we found results more consistent with that expectation. Perhaps that is why the reviewer finds them trivial. However, we view the results of this study as non-trivial because they show interesting differences from other studies. Moreover, the fact that we see more clustering of models for individual proteins in two clades (Halobacteriaceae and Thermoprotei) provides evidence that our method can reveal differential clustering.

We also note that NCBI indicates that Halobacteriaceae and Thermoprotei, our two most distinctive clades, use genetic code 11 (see link below for the translation table). Code 11 is identical to the “standard code” with the exception of having a more permissive set of start codons and a notation that UGA codes for tryptophan with low efficiency. We are actually a bit skeptical that code 11 is necessarily the best representation of translation in Halobacteriaceae and Thermoprotei - it is possible that there are code differences that are unappreciated and subtle features (e.g., the efficiency of various stop codons) is really anybody’s guess. However, we cite this to make the point that there is not obvious evidence for a very unique code in those clades (e.g., stop codon reassignments that are commonly used would be obvious in genomic data and would be noticed by NCBI). Those clades exhibit the strongest signal in our RE data, at least with respect to the clustering of models for individual proteins). We feel that this provides evidence that some of our major conclusions cannot be dismissed based on shifts in the structure of the genetic code.

Link to standard genetic code:

https://www.ncbi.nlm.nih.gov/Taxonomy/taxonomyhome.html/index.cgi?chapter=cgencodes#SG1

Link to genetic code 11:

https://www.ncbi.nlm.nih.gov/Taxonomy/taxonomyhome.html/index.cgi?chapter=cgencodes#SG11

Regarding the reviewer’s comment that “using general substitution models of protein evolution (where each model is based on the evolution of many different proteins) that combine different evolutionary signatures can result in a poor signal of the evolutionary histories” we want to reiterate that our primary goal was not tree estimation.

***

Minor comments

At the beginning of the introduction I missed at least a reference, i.e. a review, about the meaning and types of substitution models of evolution that could be useful for non-experts. I remember the following, https://www.ncbi.nlm.nih.gov/pmc/articles/PMC4620419/

***

RESPONSE:

This is an excellent suggestion. The citation that the reviewer recommended (Arenas, M. Trends in substitution models of molecular evolution. Front. Genet. 2015, 6, 319, doi:10.3389/fgene.2015.00319) is appropriate for this so we have added it.

***

The following sentence would require support by a some references “Evidence that REs exhibit a historical signal (i.e., patterns in the data that are congruent with at least some parts of the tree of life), an ecological signal (i.e., strong differences between RE estimates for taxa living in distinct environments, like thermophiles versus mesophiles), or both signals would indicate the violations of local time-reversibility are not especially problematic.”

***

RESPONSE:

This is an excellent suggestion. The citation that the reviewer recommended (Arenas, M. Trends in substitution models of molecular evolution. Front. Genet. 2015, 6, 319, doi:10.3389/fgene.2015.00319) is appropriate for this so we have added it.

***

I think the manuscript should be more cautious in many categorical sentences. For example, at end of the discussion “Moreover, a fundamental problem for all efforts to improve the realism of models of sequence evolution is that they may require the addition of free parameters and it is possible that some of those parameters will be non-identifiable (or only weakly identifiable; see Ponciano et al. [66]  for discussion).” Also it is possible to consider identifiable parameters that can be used to improve the fitting with the data (i.e., see refs in the second major comment). So that sentence is confusing and sounds biased because the opposite is also possible but not mentioned.

***

RESPONSE:

We feel that the reviewer misunderstood our point. We were not asserting that the GTR20 model had non-identifiable parameters; we were asserting that at least some models of protein evolution that *seem* reasonable might have non-identifiable parameters. It would seem wise to avoid those models in future studies (although we note that some models with non-identifiable parameters can be useful in at least some contexts, so even that statement is debateable).

We suspect that GTR20 parameters are identifiable, although we are unaware of a proof demonstrating that this is the case. Nevertheless, given that this is a very minor point with the potential to be confusing we have removed the statement.

***

Round 2

Reviewer 3 Report

I found this new version of the study more clear and detailed for readers, it properly considered some of my previous comments (I also appreciated the detailed answers from the authors) and, in all, I think the study has sufficient merits to be published. I do not have additional comments and thus I recommend accept it for publication.